# Single-cell transcriptomics unveils molecular signatures of neuronal vulnerability in a mouse model of prion disease that overlap with Alzheimer's disease

Jessy A. Slota [1,2], Lise Lamoureux[1], Kathy L. Frost[1], Babu V. Sajesh [1] & Stephanie A. Booth [1,2] ✉

Understanding why certain neurons are more sensitive to dysfunction and death caused by misfolded proteins could provide therapeutically relevant insights into neurodegenerative disorders. Here, we harnessed single-cell transcriptomics to examine live neurons isolated from prion-infected female mice, aiming to identify and characterize prion-vulnerable neuronal subsets. Our analysis revealed distinct transcriptional responses across neuronal subsets, with a consistent pathway-level depletion of synaptic gene expression in damage-vulnerable neurons. By scoring neuronal damage based on the magnitude of depleted synaptic gene expression, we identified a diverse spectrum of prion-vulnerable glutamatergic, GABAergic, and medium spiny neurons. Comparison between prion-vulnerable and resistant neurons highlighted baseline gene expression differences that could influence neuronal vulnerability. For instance, the neuroprotective cold-shock protein *Rbm3* exhibited higher baseline gene expression in prion-resistant neurons and was robustly upregulated across diverse neuronal classes upon prion infection. We also identified vulnerability-correlated transcripts that overlapped between prion and Alzheimer's disease. Our findings not only demonstrate the potential of single-cell transcriptomics to identify damage-vulnerable neurons, but also provide molecular insights into neuronal vulnerability and highlight commonalties across neurodegenerative disorders.

Prion diseases are a group of fatal and incurable neurodegenerative disorders caused by the accumulation of infectious misfolded prion proteins (PrP^Sc) in the brain, leading to a rapid clinical decline[1]. Prion replication and deposition are intricately linked with neuropathological changes, such as vacuolation, reactive gliosis, and neurotoxicity[2]. The molecular basis for prion neurotoxicity and the specific neuronal subtypes vulnerable to damage remain unclear, yet could have important therapeutic implications. This is partly owing to research efforts being complicated by selective neuronal vulnerability, a phenomenon in which different functional and spatially distinct subpopulations of neurons exhibit varying degrees of susceptibility to the cell damage and death that accompany protein misfolding[3–5]. Uncovering the reasons why certain brain regions and cell types resist disease could guide future clinical studies.

[1]Mycobacteriology, Vector-Borne and Prion Diseases Division, National Microbiology Laboratory, Public Health Agency of Canada, Winnipeg, MB, Canada. [2]Department of Medical Microbiology and Infectious Diseases, Faculty of Health Sciences, University of Manitoba, Winnipeg, MB, Canada. ✉e-mail: stephanie.booth@phac-aspc.gc.ca

Histological examination of brain tissues has identified groups of functionally distinct neurons that appear to be highly vulnerable in both human disease and rodent models. These include a subset of Pvalb[+] GABAergic neurons identified in the late '90's by Guentchev et al.[6–9] Detailed pathological characterizations show that prior to death, prion-vulnerable neurons undergo synaptic bouton degeneration[10,11], decreased dendritic spine density[12–16] and dendritic atrophy[17], which all underlie disruptions in synaptic signaling[18–23]. However, there is limited knowledge of which molecular events link PrP[Sc] accumulation with neurotoxicity and synaptic pathology. This question is complex, partly owing to the intricacy of neuronal regulatory networks that are linked by synapses on fine processes such as dendrites, which locally regulate gene and protein expression. Thus, the aim of recent studies using gene and protein expression is to link prion-induced pathological outcomes to cellular and molecular changes. While the inflammatory response of reactive glia is easily detected by genome-wide transcriptomics, neuronal transcriptional changes have been challenging to unravel[24–29]. The average of gene expression of broad neuronal populations in prion disease has been analyzed through microdissection of spatially distinct cell bodies[30–32] and by ribo-tagging[33–35] combined with RNA sequencing. However, these approaches are not capable of resolving individual neurons and chiefly identify relatively minor neuronal transcriptional changes at late clinical stages of the disease.

In principle, comparing actively degenerating neurons and those that exhibit resistance to degeneration could provide molecular insights into the basis of selective vulnerability. We assume that at the clinical endpoint of murine prion disease, neurons comprise a heterogeneous mixture of subsets with varying levels of vulnerability. This necessitates in-depth characterizations of individual degenerating neurons, which is now possible through recent advances in single-cell RNA sequencing (scRNAseq)[36–38]. Our group[39] and others[40], have begun to harness these techniques to study prion pathophysiology and have pinpointed transcriptional changes among neurons and glia. While these initial datasets have not yet been leveraged to characterize neuronal vulnerability, single-cell technologies are well poised for this purpose. Our group aims to apply scRNAseq to live brain cells isolated from prion-infected mice. However, isolating intact live neurons from adult mice is extremely challenging due to their complex interactions via multiple cellular processes and projections. Thus, the number of neurons in our previous scRNAseq dataset was too small to discern their relative vulnerabilities[39]. Nevertheless, successfully captured intact neurons contain more complete transcriptomes than nuclei. Improvement to our methodology and isolation of additional live neurons should enable robust characterizations of neuronal vulnerability in prion disease.

In the present study, we performed further single-cell transcriptomics of live brain cells isolated from prion-infected mice at the onset of clinical signs, this time implementing cell isolation methods aimed at enriching neurons. Analysis of the resulting scRNAseq data revealed prominent transcriptional responses within the cortical glutamatergic, nucleus accumbens GABAergic and striatal medium spiny neurons. We found that degenerating neuronal populations consistently exhibit a net depletion of synaptic gene expression. This was true across multiple datasets from studies of both prion and Alzheimer's disease, irrespective of the –omics technology used. With this in mind, we developed a scoring system to correlate the extent of synaptic gene depletion with neuronal damage, thereby ranking neurons in our dataset based on their vulnerability to degeneration. This analysis showed striatal medium spiny neurons as particularly vulnerable to prion disease, whereas hippocampal glutamatergic neurons were relatively resistant. In addition, we identified baseline transcriptional differences between neuronal subsets classified as vulnerable and resistant, which might influence vulnerability. Meta-analysis of published datasets showed that these vulnerability-correlated gene signatures partially overlapped between prion and Alzheimer's disease. We also identified *Rbm3*, a cold-shock protein that protects against prion disease[41,42], as highly expressed by prion-resistant hippocampal neurons and upregulated in prion disease.

## Results

### A single-cell transcriptional atlas of live neurons in prion disease

We previously created a live single-cell transcriptional atlas of murine prion disease[39], although the final proportion of mature neurons was less than 3% of the total single cells sequenced. To increase the yield of neurons in the current study, we aimed to deplete non-neuronal cells from dissociated brain tissues prior to scRNAseq. We used prion-infected (*n* = 5) and mock-infected (*n* = 4) mice and collected brains during clinical disease, characterized by reduced body condition, kyphosis, clasping during tail suspension, and ataxia. Forebrain and midbrain tissues were collected and immediately dissociated into live single-cell suspensions. Non-neuronal cells were depleted using magnetic beads coated with antibodies specific for glia, oligodendrocytes, endothelial cells, and fibroblasts, and the remaining cells were immediately used for droplet-based single-cell RNA sequencing. In total, 21 droplet-based single-cell RNA sequencing libraries were pre-processed, integrated, and clustered as described previously[39]. The resulting 100,946 high-quality transcriptomes were classified into 34 clusters of glutamatergic, GABAergic, medium spiny, and immature neurons, endothelial cells, and microglia (Fig. 1a, b and Supplementary Fig. 1). The abundance of endothelial cells and immature neurons suggested inefficient depletion of these cell types. However, the overall cell-type composition was similar between prion- and mock-infected mice (Fig. 1c), indicating the consistency of the methodology. Furthermore, differential gene expression analysis revealed prion-associated transcriptional differences (FDR < 0.05) amongst the major brain cell types captured, including neurons, endothelial cells, *Dcx*[+] immature neurons, microglia, and choroid plexus cells (Fig. 1d–g). As in our previous study[39], these prion-altered transcripts reflect the canonical signatures of prion pathophysiology, verifying this scRNAseq atlas as a reliable resource.

Despite the inclusion of experimental steps to deplete non-neuronal cells, isolation of neurons (~ 8%) was only marginally improved compared to our previous study (~ 3%)[39]. To increase the power for analyzing mature neurons, we integrated the transcriptomes of the 8148 mature neurons resolved in the current study with our previous dataset of 1248 and 1358 neurons isolated from the cortex and hippocampus, respectively[39]. Using Seurat's harmony dataset-integration method to preserve separation of neurons isolated from different brain regions (Supplementary Fig. 2), we produced a final integrated atlas of 10,754 neuronal transcriptomes (Fig. 2a). Sub-clusters within the neuronal atlas were annotated based on homology with the Allen Brain Atlases' high-resolution reference mouse brain cell atlas[43] (Fig. 2b, c). In total, 42 sub-clusters were resolved, comprising predominantly of glutamatergic neurons from the cortex, hippocampus, and anterior olfactory nucleus, GABAergic neurons from the nucleus accumbens and/or olfactory bulb, striatal medium spiny neurons, and a smaller number of *Pvalb*[+], *Sst*[+], *Sncg*[+], and *Lamp5*[+] GABAergic neurons. Expression of neuronal marker genes confirmed sub-cluster identity (Fig. 2d and Supplementary Fig. 3). Glutamatergic neuronal classes were distinguished by the expression of *Slc17a7*, *Cck*, and *C1ql3*, whereas GABAergic neuronal subsets were demarked by *Gad1*, *Gad2*, *Slc32a1*, *Cpa6*, *Drd1* and *Drd2*. Several neuronal sub-clusters were annotated as mixed because they comprised a heterogeneous mixture of neuronal subclasses and tended to have lower expression of relevant neuronal marker transcripts. We postulated that mixed neuronal sub-clusters may contain lower-quality transcriptomes or included cell debris due to incomplete dissociation and were, therefore, not the focus of our subsequent analyses.

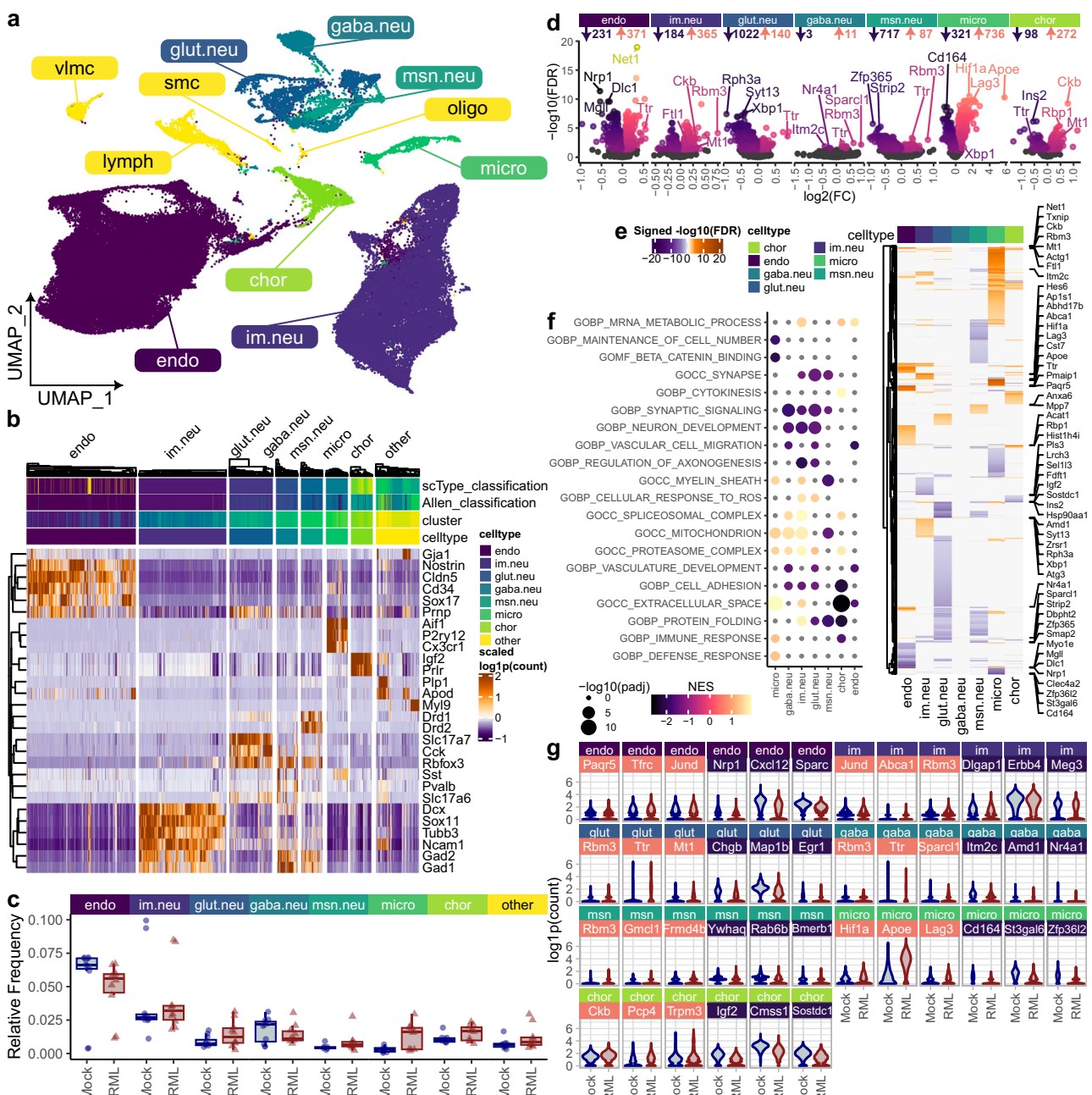

**Fig. 1 | A neuron-enriched single-cell transcriptional atlas of prion disease.**
**a** UMAP plot of 100,946 cells classified into 34 clusters of endothelial cells,
immature neurons, glutamatergic, GABAergic, and medium spiny neurons,
and microglia, and choroid plexus cells, among others. **b** Hierarchical clustering
(Pearson correlation) of gene expression profiles was used to visualize the speci-
ficity of marker transcript (identified with Seurat's Wilcoxon rank-sum test)
expression across different brain cell types. **c** Relative frequencies of major brain
cell types were compared between RML (*n* = 5) and Mock (*n* = 4) infected mice using
scCODA's Bayesian model that uses a direct posterior probability approach for FDR

estimation (* FDR < 0.05). Boxplots show mean (center line), 25th and 75th percen-
tiles (upper and lower hinges), and 1.5*IQR (whiskers). **d** volcano plots and (**e**)
heatmap show prion-altered transcripts amongst each major brain cell type.
MAST's likelihood-ratio test was used for differential expression analysis (FDR
*p-value* < 0.05). **f** Plot shows top pathways enriched with prion-altered transcripts
among major brain cell types, as determined via pre-ranked gene set enrichment
analysis using fgsea's adaptive multi-level split Monte-Carlo scheme for *p*-value
estimation. **g** Violin plots show prion-altered (identified with MAST's likelihood-
ratio test) expression of select transcripts within each brain cell type.

## Single-cell RNA sequencing resolves comparable neuronal sub-populations in prion and mock-infected mice

We reasoned that cell death of neuronal subsets particularly vulner-
able to prion neurotoxicity might lead to their depletion compared to
the equivalent subset isolated from mock-infected animals. Therefore,
we examined the relative frequency of each neuronal sub-cluster using
scCODA's Bayesian model and identified relatively minor changes

among neuronal subsets isolated from the forebrain and midbrain, but
not amongst neurons that originated from bulk dissected cortical or
hippocampal tissues (Fig. 2e and Supplementary Fig. 4). Sub-clusters
1.Mixed.Glut and 15.Mixed.Glut elicited prion-associated increases in
relative frequency. These sub-clusters likely correspond to low-quality
damaged neurons because they comprised a mixture of cortical and
hippocampal glutamatergic neurons (Fig. 2c), yet tended to only

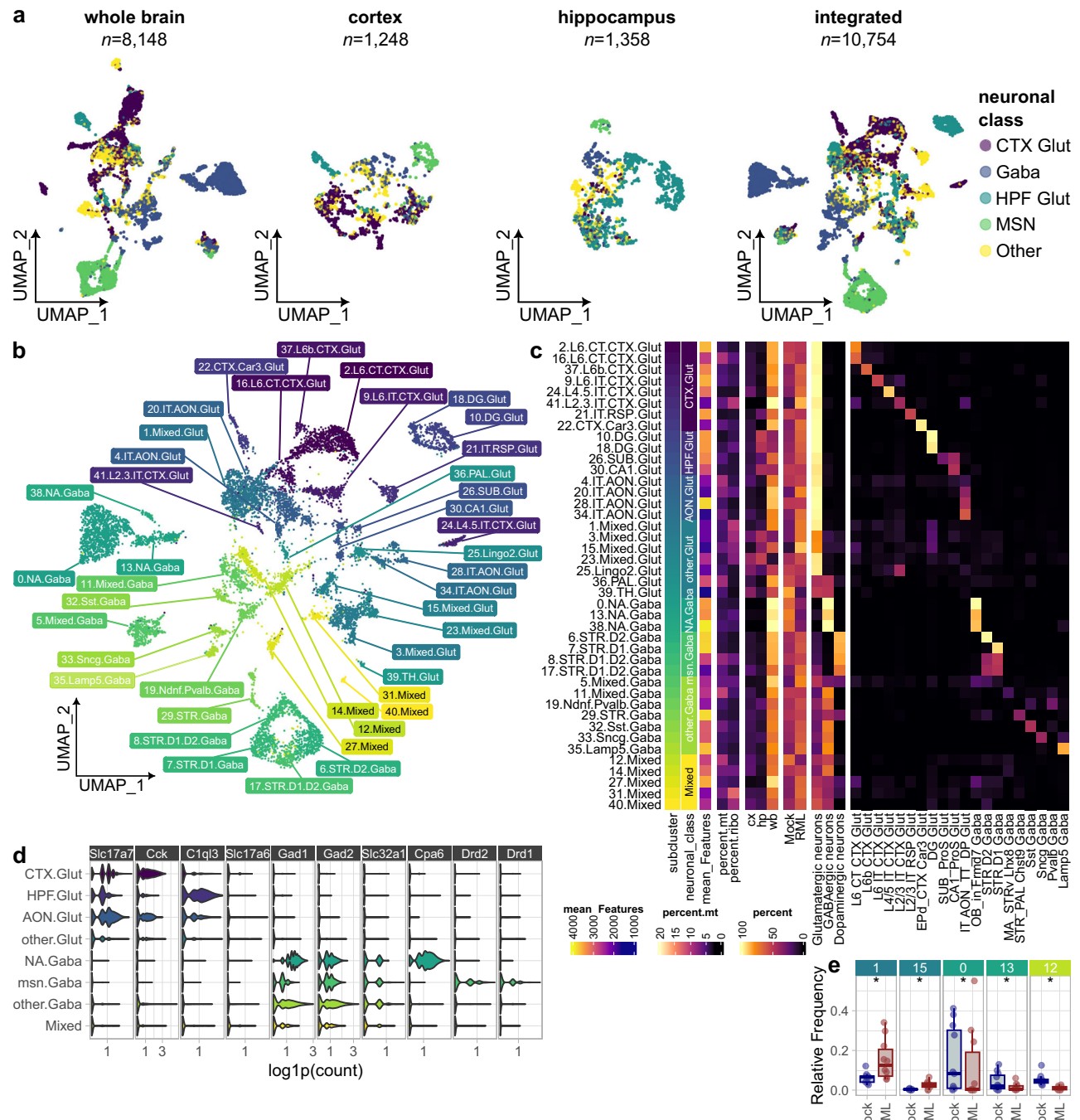

**Fig. 2 | Dataset integration produces a transcriptional atlas of high-quality live neurons during prion infection. a** UMAP plots show neuronal transcriptomes originating from scRNAseq datasets made from the whole brain (this study), cortex (Slota et al. 2022), and hippocampus (Slota et al. 2022), which were integrated to produce a single-cell transcriptional atlas of prion-infected neurons. **b** UMAP plot of 10,754 glutamatergic, GABAergic, and medium spiny neurons classified into 42 sub-clusters. **c** Each sub-cluster was annotated with its neuronal class, mean genes detected per cell, mean percent mitochondrial and ribosomal transcript expression, percentage of cells originating from cortical, hippocampal, and whole brain tissues, percentage of cells originating from RML and Mock treated mice, and percentage of cells classified according to scType and Allen brain atlas. **d** Violin plots show expression of transcripts that demarked overarching neuronal sub-classes (identified with Seurat's Wilcoxon rank-sum test). **e** Credible differences in cell composition between RML (*n* = 5) and Mock (*n* = 4) infected mice among whole brain samples were identified using scCODA's Bayesian model (* FDR < 0.05, sub-cluster 33.Sncg.Gaba was automatically selected as the reference). Boxplots show mean (center line), 25th and 75th percentiles (upper and lower hinges), and 1.5*IQR (whiskers).

weakly express glutamatergic neuronal markers (Supplementary Fig. 3). Possibly, this prion-associated increase in neuronal clusters with transcriptomes of lower quality could indicate that the cells were damaged and more sensitive to stress and shearing during cell isolation. We also detected a decrease of sub-clusters 0.NA.Gaba and 13.NA.Gaba in cells isolated from prion-infected animals, indicating

potential vulnerability. Given the small number of neurons in our study, it is difficult to interpret whether these differences are related to neuronal loss, especially given the technical challenges of isolating mature neurons and the imprecision of bulk tissue dissection. However, the overwhelmingly similar neuronal composition between prion and mock-infected mice (Supplementary Fig. 4b) confirms that our

atlas is a robust starting point to examine neuronal transcriptional variation in prion infection.

## Prion disease elicits distinct transcriptome responses between neuronal sub-classes

Prion-induced changes in neuronal gene expression are expected to coincide with neuronal damage, raising the possibility of leveraging gene expression to distinguish degenerating neuronal subsets. Therefore, identifying transcripts that elicit consistent gene expression changes across diverse neuronal classes in prion disease would be highly valuable. On the other hand, we anticipated that the heterogeneity of neuronal gene expression would minimize the number of transcripts that can be detected across all neuronal sub-clusters. To probe for overlapping prion-altered transcripts, we first identified differentially expressed genes (DEGs) for each neuronal sub-cluster by comparing cells isolated from prion and mock-infected mice using MAST's two-part Hurdle model (FDR < 0.05; Fig. 3a and Supplementary Fig. 5a). This revealed prominent transcriptional responses of cortical glutamatergic neurons, nucleus accumbens GABAergic neurons and striatal medium spiny neurons. In total, we identified 385 decreased transcript enriched in ontologies related to vesicle-transport, TORC1 signaling, and cholesterol biosynthesis, and 61 increased transcripts related to oxidative stress, nucleophagy, and cytoplasmic transport (Fig. 3c). To help visualize the effect sizes associated with these prion-associated transcriptional changes, we provide violin plots of top differentially expressed genes per sub-cluster in Fig. 3f.

As expected, prion-altered transcriptional signatures were diverse across different neuronal classes, reflected by the minimal overlap of prion-altered transcripts between neuronal sub-clusters (Fig. 3b, d). To account for differences in statistical power between neuronal sub-clusters, we also evaluated whether trends in prion-altered gene expression were similar between neuronal sub-clusters through correlation analysis of signed −log10(FDR) values (Fig. 3e). We observed strong within-class correlation amongst sub-clusters of cortical glutamatergic neurons, AON glutamatergic neurons, NA GABAergic neurons, and medium spiny neurons. Conversely, between-class correlation among sub-clusters was relatively low. This implies that prion-altered transcription is similar between neuronal sub-clusters within the same class, whereas divergent neuronal classes elicit distinct transcriptional responses.

Despite the minimal overlap of prion-altered neuronal gene expression, we identified a small number of genes that elicited consistent prion-altered expression changes across multiple neuronal sub-clusters (Supplementary Fig. 5b). *Rbm3* and *Ttr* showed a prion-associated trend of increased expression across almost every neuronal sub-cluster resolved, achieving statistical significance in several instances. Neuronal *Rbm3* upregulation is noteworthy since the cold shock response protein Rbm3 has already been established as neuroprotective in prion-infected mice[41,42]. Genes that were the most consistently downregulated across multiple neuronal sub-clusters were *Rab6b*, *Egr1*, *Rock2*, and *Ier5*. Downregulation of the immediate early genes *Egr1* and *Ier5*, plus several others (e.g., *Egr3*, *Homer1*, *Arc*), suggests loss of synaptic plasticity. Other than these few genes, relatively few transcripts were altered across multiple sub-clusters, indicating the specificity of prion-altered neuronal transcription across distinct neuronal subsets.

## RNA-FISH validates the prion-altered abundance of neuronal *Rbm3*, *Egr1*, and *Rab6b*

To verify differential gene expression, we employed RNA fluorescence in situ hybridization (FISH) to determine the abundance of select transcripts across spatially-resolved neuronal subsets in prion disease. We examined *Ttr* and *Rbm3* because they were consistently upregulated across multiple neuronal clusters, and *Egr1* and *Rab6b*, which were downregulated in cortical glutamatergic and striatal medium

spiny neurons, respectively. Towards this end, six panels of RNA FISH probes were used to measure *Ttr*, *Rbm3*, and *Egr1*, expression in Glutamatergic and GABAergic neurons, and *Ttr*, *Rbm3*, and *Rab6b* expression in medium spiny neurons (Fig. 4 and Supplementary Fig. 6). We collected region of interest (ROI) images targeting the cortex, CA1, and dentate gyrus regions to assess Glutamatergic and GABAergic neurons, and the striatum for medium spiny neurons. This approach confirmed the striking prion-associated increased expression of *Rbm3* across several distinct types of neurons (Fig. 4g). Furthermore, *Egr1* was strongly downregulated in cortical glutamatergic neurons and GABAergic neurons. We also confirmed the prion-associated downregulation of *Rab6b* in both *Drd1*[+] and *Drd2*[+] medium spiny neurons. A trend of increased *Ttr* expression was also detected across the different neuronal classes investigated; however, this trend did not achieve significance. By leveraging previously published bulk RNAseq datasets[24,30,33,44], we also examined the prion-altered abundance of these genes longitudinally in prion-infected mice (Fig. 4h). *Rbm3* upregulation combined with decreased *Egr1* and *Rab6b* was readily detected at the clinical endpoint of prion disease, but not at earlier timepoints. We, therefore, restricted our RNA-FISH validation to the clinical endpoint of prion disease. Overall, these RNA-FISH findings closely resembled the gene expression changes detected within our neuronal single-cell atlas, confirming the robustness of the dataset.

## Pathway-level depletion of synapse-related gene expression demarks degenerating neurons in prion disease

To distinguish prion-vulnerable neurons, we sought to identify a universal transcriptional marker of neuronal damage. Other than *Rbm3*, the expression of individual genes was variable across different neuronal subsets. We reasoned that analyzing groups of functionally related genes might be more powerful compared to individual transcript analysis. The major signature of prion-induced neuronal damage is expected to reflect synaptic pathology[30,39]. To illustrate this point, gene set enrichment analysis (GSEA) was performed on previously published bulk RNAseq datasets[24,30,33,44] from mouse models of prion infection (Fig. 5a), confirming the decrease in genes related to synaptic signaling and neuronal damage commonly observed at the endpoint of murine prion disease. However, bulk transcriptional signatures cannot distinguish between cell composition changes (i.e., neuronal loss and glial proliferation), or gross expression changes in cell types within tissues.

To examine whether prion infection results in the depletion of synapse-related gene expression within individual neurons, we performed GSEA on the lists of differentially expressed transcripts from each neuronal sub-cluster (Fig. 5b and Supplementary Fig. 7a). Examining the top prion-altered pathways confirmed that synapse-related gene expression was indeed depleted across different classes of neurons in prion-infection. Interestingly, we identified pathways related to vacuoles, proteolysis, and protein folding that were increased in a few neuronal sub-clusters, perhaps reflecting a stress response. As an additional means of verifying which sub-clusters elicited a decrease in synaptic gene expression, we also performed GSEA on gene expression profiles of individual neurons, comparing single-cell synapse enrichment scores between prion and mock-infected mice (Fig. 5c and Supplementary Figs. 7b, 8). Overall, this analysis revealed disease-associated decreases in synaptic gene expression across vulnerable sub-clusters of glutamatergic, medium spiny, and NA GABAergic neurons, likely indicative of neurite and synaptic damage.

We next identified individual prion-altered transcripts that were localized to synapses using SynGO (Fig. 5d). Numerous transcripts were localized to both the pre- and post-synaptic compartments, and a number are expressed within synaptic vesicles. We examined the effect size of these synapse-localized genes across neuronal sub-clusters (Fig. 5e). This confirmed that synapse-related transcripts were

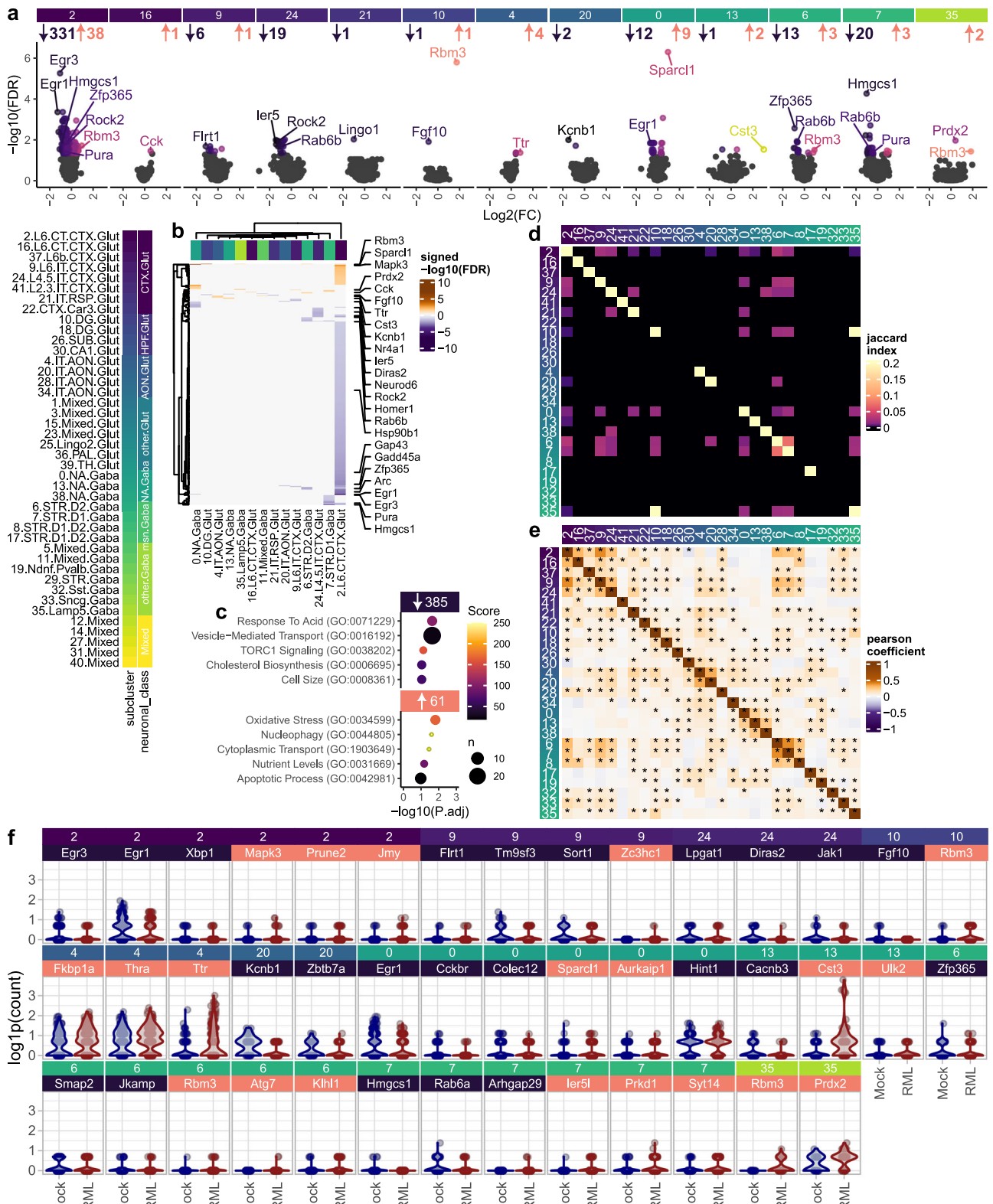

**Fig. 3 | Prion-altered neuronal gene expression.** Prion-altered transcripts were identified among high-quality neuronal clusters using MAST's generalized linear mixed model with a random effect for individual libraries. Prion-altered transcription was visualized with (**a**) volcano plots and (**b**) hierarchical clustering. **c** Biological process gene ontologies enriched with all prion-altered genes that were increased and decreased among pure neuronal clusters were determined using the Fisher exact test implemented by Enrichr. Heatmaps show (**d**) overlap and (**e**) correlation between differential expression results for each neuronal subcluster. **f** Violin plots show expression of up to the top 3 prion-altered transcripts per neuronal cluster (identified with MAST's likelihood-ratio test).

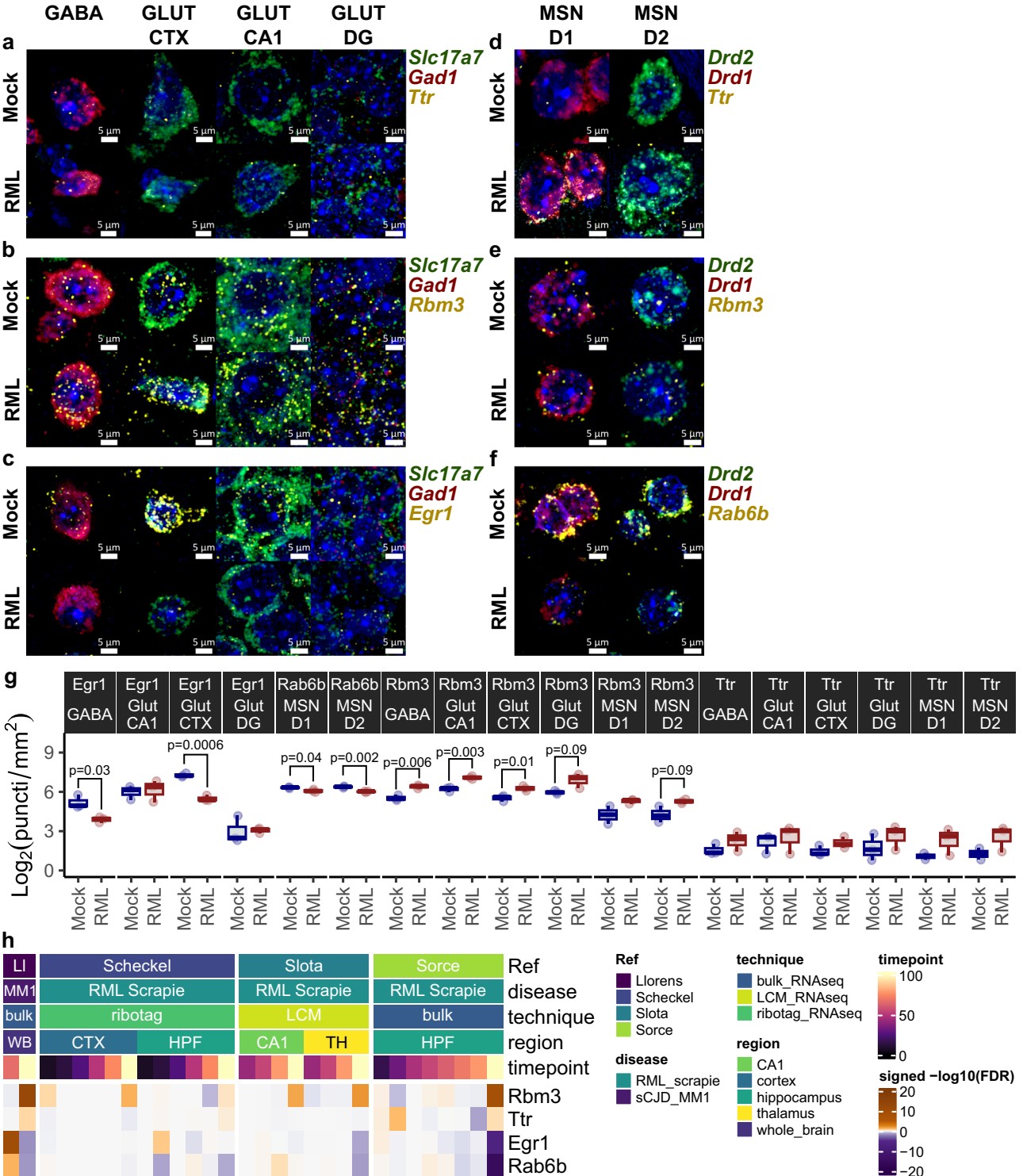

**Fig. 4 | RNA-FISH validation of prion-altered neuronal gene expression.** Representative images show RNAscope probe panels that were used to assess (**a**) *Ttr*, (**b**) *Rbm3*, and (**c**) *Egr1* expression in *Slc17a7*⁺ glutamatergic neurons and *Gad1*⁺ GABAergic neurons in the cortex, CA1, and dentate gyrus. Three additional RNAscope probe panels assessed (**d**) *Ttr*, (**e**) *Rbm3*, and (**f**) *Rab6b* expression in *Drd1*⁺ and *Drd2*⁺ striatal medium spiny neurons. Scale bar = 5 µm. **g** Gene expression was measured as the average puncti density per mouse and compared between prion (*n = 3*) and mock-infected (*n = 3*) mice using the student's *t* test. *P*-values: 0.033, 0.79, 0.00064, 0.91, 0.041, 0.0018, 0.0060, 0.0031, 0.013, 0.095, 0.11, 0.094, 0.25, 0.65, 0.13, 0.32, 0.17, 0.14. Boxplots show mean (center line), 25ᵗʰ and 75ᵗʰ percentiles (upper and lower hinges), and 1.5*IQR (whiskers). **h** Prion-altered abundance, expressed as the signed −log₁₀(FDR), of *Ttr*, *Rbm3*, *Egr1*, and *Rab6b*, was examined longitudinally in previously published bulk RNAseq datasets of prion-infected mice.

predominantly decreased within prion-infected neurons, with only a few eliciting an increase (e.g., *Sparcl1*, *Mapk3*, and *Prune2*). Sub-clusters of cortical glutamatergic, NA GABAergic, and striatal medium spiny neurons elicited the strongest decrease in synapse pathways via GSEA

and also elicited the strongest depletion of individual synaptic genes. Collectively, these data emphasize the consistency of the loss of neuronal gene expression across brain regions and neuronal classes, presumably elicited by prion neurotoxicity.

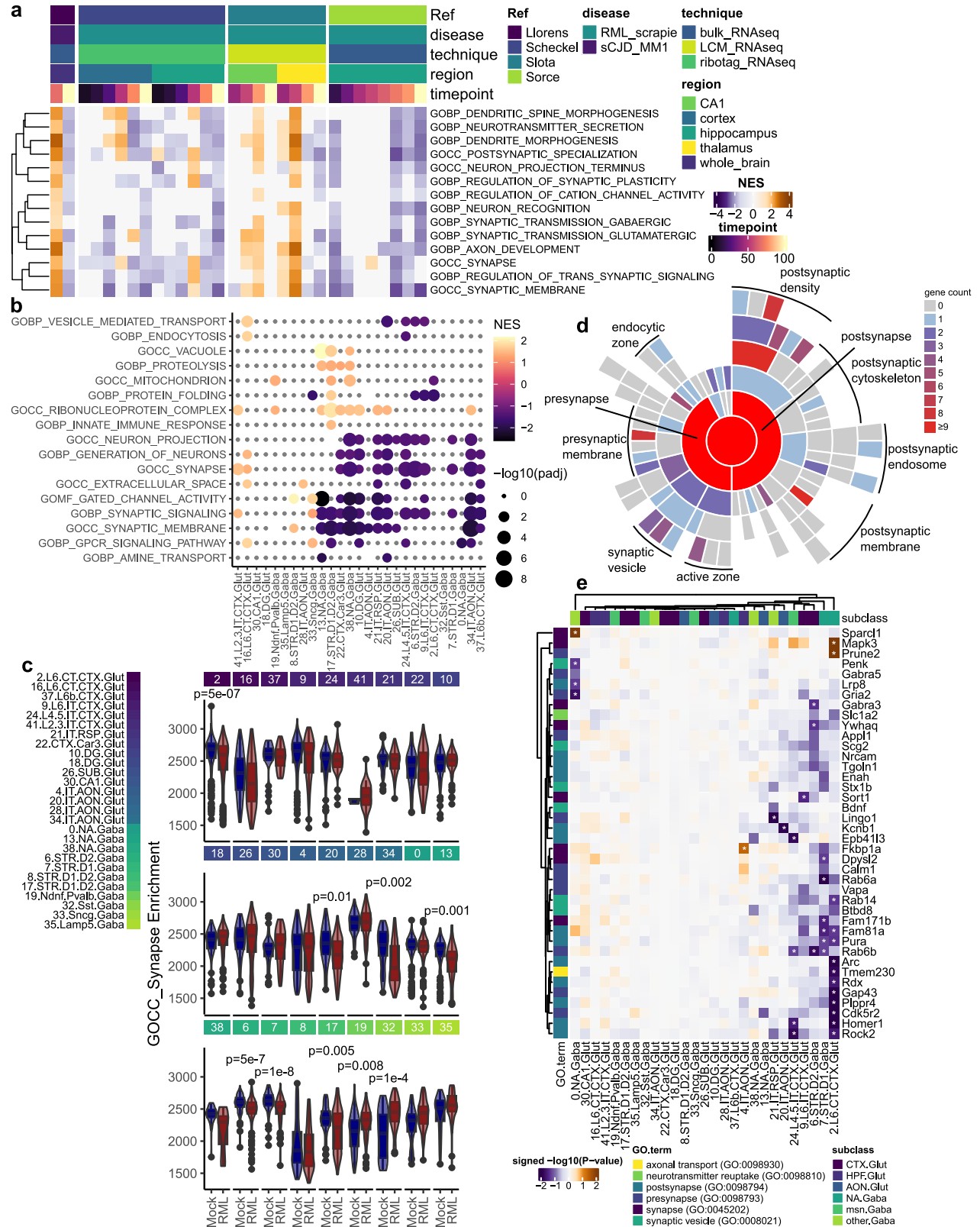

### Scoring neuronal vulnerability reveals medium spiny neurons as especially vulnerable to prion-induced damage

Given that loss of neuronal identity and synaptic signaling is the predominant molecular signature of degenerating neurons in prion disease, we hypothesized that this signature could be used to identify and rank vulnerable neuronal subsets. We assigned neuronal sub-clusters a vulnerability score (Fig. 6a, c) that was defined as the

−signed(−log10(adjusted-$p$-value)) of the GOCC-Synapse gene set from pre-ranked GSEA. Thus, these vulnerability scores take advantage of the established signature of neuronal degeneration (Fig. 5), reflecting the effect size of prion-induced synapse gene depletion within each neuronal sub-cluster. Neuronal sub-clusters with a vulnerability score greater than 1 were classified as vulnerable (i.e., GOCC-Synapse GSEA adjusted $p$-value < 0.1), while the remaining

**Fig. 5 | Neuronal synaptic gene expression is consistently depleted in prion disease. a** GSEA, using fgsea's adaptive multi-level split Monte-Carlo scheme for *p*-value estimation, was used to compare the enrichment of synapse-related gene sets throughout prion disease progression from previously published bulk RNAseq datasets of prion-infected mice. **b** Fgsea's adaptive multi-level split Monte-Carlo scheme for *p*-value estimation was used to identify gene sets that were enriched with prion-altered transcripts (pre-ranked using MAST's likelihood ratio rest) within each neuronal cluster. **c** Enrichment of the GOCC Synapse gene set was measured

within individual neurons and compared between prion (*n* = 3) and mock (*n* = 3) infected mice using the *t* test and FDR correction implemented by Escape (*p*-values: 5.1E-07, 0.38, 1, 1, 0.89, 1, 1, 1, 0.29, 1, 1, 1, 0.012, 1, 0.0025, 0.10, 0.0010, 0.21, 5.2E-07, 1.4E-08, 1, 0.0053, 0.0075, 0.00013, 0.034, 1). Boxplots show mean (center line), 25th and 75th percentiles (upper and lower hinges), and 1.5*IQR (whiskers). **d** Sunburst plots show synaptic localization of prion-altered transcripts (number of genes). **e** Heatmap shows the effect size of synapse-related transcripts within each neuronal subcluster (MAST's likelihood ratio test; * FDR corrected *p*-value < 0.05).

---

neuronal sub-clusters were classified as resistant (Fig. 6b, c). A greater proportion of neurons from the atlas were classified as resistant rather than vulnerable, however vulnerability scores were distributed similarly among neurons isolated from prion and mock-infected mice (Fig. 6d, e). Hippocampal glutamatergic neuronal sub-clusters tended to have lower vulnerability scores, with sub-cluster 10.DG.Glut only meets the criteria for classification as vulnerable. Thus, hippocampal glutamatergic neurons appeared to be relatively resistant compared to the other types of neurons captured by our atlas. Conversely, striatal medium spiny neurons were ranked as the most vulnerable to prion infection.

We also independently assessed neuronal vulnerability in our prion mouse model using RNA FISH. Three probe panels were used to identify glutamatergic (*Slc17a7*, *Slc17a6*, and *Grin2b*), GABAergic (*Gad1*, *Pvalb*, and *Sst*), and medium spiny neurons (*Gad1* and *Drd1*) in prion- (*n* = 3) and mock-infected (*n* = 3) mice (Fig. 6f–h). ROI images were collected of the striatum for medium spiny neurons, and of the cortex, hippocampus, midbrain, and thalamus for GABAergic and Glutamatergic neurons (Supplementary Fig. 9). Neurons were counted within each ROI image, and we assessed the vulnerability of each neuronal subset based on their prion-associated depletion (Fig. 6i). When examining neuronal depletion, the thalamic GABAergic and glutamatergic neurons captured by our RNA-FISH dataset appeared to be the most vulnerable to prion disease (Fig. 6i). Thalamic neurons were absent from the scRNAseq dataset, perhaps because their vulnerability made them especially sensitive to our cell isolation protocol. We also were unable to assess AON or NA neurons in our RNA-FISH dataset because these regions were absent from the tissue blocks that were available for sectioning and staining. Nevertheless, RNA-FISH validated that striatal medium spiny neurons are especially vulnerable to prion infection, in agreement with our scRNAseq-based ranking of neuronal vulnerability (Fig. 6a).

### Correlating baseline gene expression with neuronal vulnerability scores reveals transcriptional markers of vulnerable and resistant neurons in prion disease

Having ranked neuronal sub-clusters by their vulnerability score, we next leveraged our dataset to interrogate the relationship between baseline neuronal gene expression with predicted vulnerability to prion-induced damage. In this way, we sought to identify biomarkers predictive of neurons that may be more susceptible to prion disease. Neurons isolated from prion-infected mice were removed, and we correlated baseline transcriptional profiles (from mock-infected mice) with vulnerability scores through differential expression analysis with MAST's two-part hurdle model (Fig. 7a, b and Supplementary Data 5). As neurons from prion-infected mice were excluded from this analysis, vulnerability markers were not influenced by disease-associated gene expression changes. As expected, we identified numerous transcripts that demarked the neuronal sub-clusters classified as vulnerable and resistant. Among the top vulnerability-correlated transcripts were *Tubb2a*, *Ppm1e*, and *Nrip3*, expressed by resistant neurons, and *Penk*, *Foxp1,* and *Cacna2d3*, expressed by vulnerable neurons (Fig. 7c). We also examined the expression of select vulnerability-associated transcripts across all neuronal sub-clusters (Fig. 7d). This revealed the top vulnerability-associated transcript, *Penk*, as being highly expressed by

sub-cluster 6.STR.D2.Gaba, which had the highest vulnerability score (Fig. 6a). This confirmed that *Penk*+*Drd2*+ medium spiny neurons are selectively vulnerable.

Functional annotation of transcripts differentially expressed by vulnerable and resistant neuronal subsets revealed several biological processes that were selected as representative biomarker signatures (Fig. 7e). Vulnerable neurons highly expressed transcripts linked to synaptic transmission, memory, and cellular adhesion, while resistant neurons abundantly expressed genes related to vesicle trafficking, negative regulation of supramolecular fiber assembly, and ubiquitin-mediated protein catabolism. Interestingly, several vulnerability-correlated transcripts have been previously linked with prion disease. For instance, the metabotropic glutamate receptors *Grm5* and *Grm8* mediate prion toxicity[45,46], while *Bmerb1*[47], *Macrod2*[46], and *Stmn2*[48] are implicated as putative genetic risk loci for prion disease. Several tubulin-encoding genes, including *Tubb2a*, *Tubb2b*, *Tubb4a*, *Tubb4b*, and *Tubb5*, were highly expressed by prion-resistant neurons. This is of note because, in a previous study, we found PrP^C-over-expressing neuronal cultures, which were demarked by high tubulin expression, to completely resist prion infection[49]. Moreover, the baseline expression of the neuroprotective stress-response gene *Rbm3*[41,42] was higher in prion-resistant neurons (Supplementary Data 5), and this was supported by RNA FISH that detected conspicuously low levels of *Rbm3* expressed by selectively vulnerable medium spiny neurons (Fig. 4g). The prion-associated upregulation of *Rbm3* consistently seen across different neuronal subsets (e.g., Figs. 3, 4) could imply a protective response. Thus, it is tempting to speculate that neuronal expression of specific genes could influence vulnerability to prion-induced damage. However, no other scRNAseq datasets for prion disease are currently available to test this hypothesis.

### Vulnerability-correlated transcripts partially overlap between prion and Alzheimer's disease

To determine whether our approach to identifying vulnerability-correlated transcripts would produce similar results when applied to a different dataset, we performed a meta-analysis using single-nucleus RNAseq datasets of Alzheimer's disease patients previously published by Grubman et al.[50] and Leng et al.[36]. We first used MAST to identify transcripts differentially expressed between AD cases versus controls in the Grubman dataset (Supplementary Fig. 10a), and between AD cases based on Braak score in the Leng dataset (Supplementary Fig. 10d). Next, GSEA was used to identify enriched pathways amongst the MAST differential expression results, confirming that synaptic gene expression is also consistently decreased in damage-vulnerable neuronal sub-clusters in Alzheimer's disease (Supplementary Fig. 10b, e). Thus, vulnerability scores were again computed as the −signed(-log10(FDR)) of AD-associated enrichment of the GOCC synapse gene set, which we used to rank the predicted vulnerability of each neuronal sub-cluster (Supplementary Fig. 10c, f). Finally, to correlate baseline transcriptional changes with predicted neuronal vulnerability to Alzheimer's disease, we performed differential expression analysis against vulnerability score using MAST (Fig. 8a, b). We performed differential expression analysis on neurons from control cases in the Grubman Dataset, and

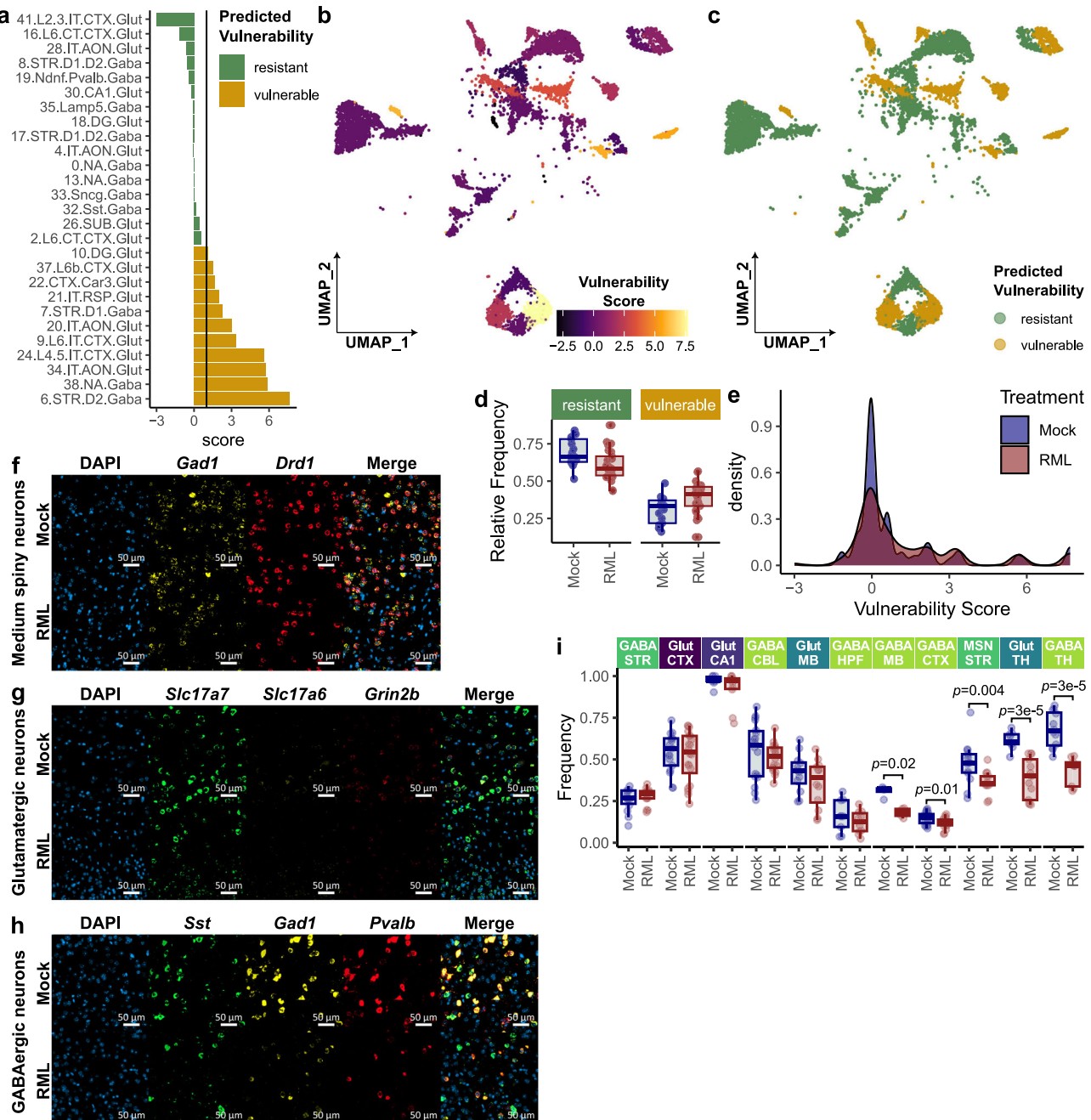

**Fig. 6 | Identification of vulnerable neuronal subsets in prion-infected mice.**
**a** Neuronal sub-clusters were ranked by vulnerability score, defined as −signed(-log10(FDR)) of prion-altered enrichment of the GOCC Synapse gene set, identified through GSEA against pre-ranked transcripts from the MAST differential expression analysis. Vulnerable neuronal clusters were defined as those with vulnerability score >1 (i.e., decreased GOCC Synapse enrichment from GSEA with FDR = 0.1). UMAP plots are shown for neurons, annotated with (**b**) vulnerability scores and (**c**) the predicted vulnerability of each neuronal cluster. **d** Cellular composition was compared between neurons defined as 'vulnerable' and 'resistant' in prion- (n = 13) and mock-infected (n = 8) mice. **e** Distribution of neuronal vulnerability scores were compared between prion and mock-infected mice. In an independent assessment of neuronal vulnerability, three panels of RNAscope probes were used to identify

Glutamatergic neurons (*Slc17a7*, *Slc17a6*, *Grid2*), GABAergic neurons (*Gad1*, *Pvalb*, *Sst*) and MSN's (*Gad1*, *Drd1*) in formalin-fixed coronal sections from RML and Mock infected mice. Regions of interest (ROI's) were selected from the striatum, cortex, cerebellum, hippocampus, thalamus, and midbrain for analysis. Representative ROI's are shown for (**f**) striatal medium spiny neurons, (**g**) cortical glutamatergic neurons, and (**h**) cortical GABAergic neurons. **i** Neuronal vulnerability was assessed by comparing the abundance of each neuronal subset between prion- (n = 3) and mock-infected (n = 3) mice (two-sided Wilcoxon rank-sum test; p-values: 0.84, 0.52, 0.49, 0.48, 0.41, 0.36, 0.016, 0.012, 0.0039, 2.8E-05, 2.7E-05). Boxplots (**d** and **i**) show mean (center line), 25th and 75th percentiles (upper and lower hinges), and 1.5*IQR (whiskers).

cases with Braak Score of 0 in the Leng dataset. Vulnerability-correlated transcripts in Alzheimer's disease were enriched in gene ontologies that were similar to prion disease (Fig. 8c, d). For instance, vulnerable neurons expressed genes related to synaptic transmission and glutamate signaling, whereas resistant neurons expressed genes

related to the regulation of supramolecular fibers, neuron migration, and axonogenesis.

To identify commonalities across neurodegenerative disorders, we next identified vulnerability-correlated transcripts that overlapped between prion and Alzheimer's disease (Fig. 8e, f). The overlap of

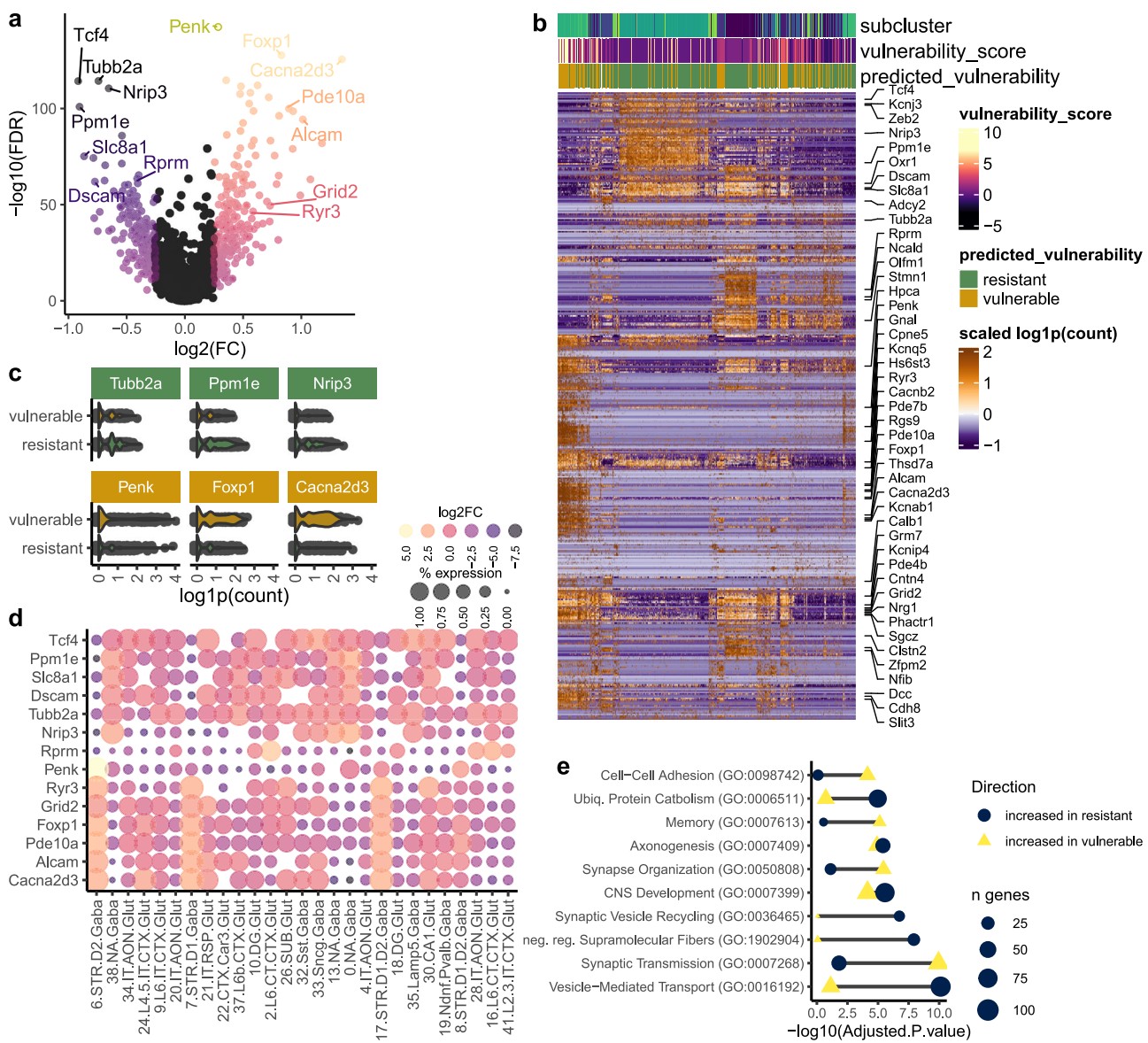

**Fig. 7 | Molecular correlates of neuronal vulnerability to synaptic damage in prion disease. a** Transcripts that correlated with neuronal vulnerability score were identified through differential expression analysis with MAST's two-part generalized linear model, assigning vulnerability score as a co-variate. **b** Hierarchical clustering of gene expression profiles of vulnerability-correlated transcripts (identified with MAST's likelihood ratio test) between neuronal clusters classified as vulnerable and resistant. **c** Violin plots show expression of the top 3 transcripts associated with vulnerable and resistant neurons (identified with MAST's likelihood ratio test). **d** Expression of select vulnerability-correlated transcripts is shown per neuronal subcluster. **e** Enrichr's Fisher exact test was used to compute gene ontologies enriched with transcripts that were significantly associated with vulnerable and resistant neurons (identified with MAST's likelihood ratio test).

vulnerability markers between all three datasets (this study, Grubman, and Leng) was surprisingly high – 119 and 61 transcripts consistently demarked vulnerable and resistant neurons, respectively (Fig. 8g). Counterintuitively, we noted that *Prnp* was among the markers of resistant neurons common to prion and Alzheimer's disease. The 119 transcripts expressed by vulnerable neurons were related to synaptic transmission, glutamate signaling, and cation channels, whereas the 61 transcripts expressed by resistant neurons were related to synapse assembly and negative regulation of amyloid beta formation (Fig. 8h). We visualized the magnitude of expression differences between resistant and vulnerable neurons for three of the more promising markers of resistant (*Erbb4*, *Tcf4*, and *Adarb2*) and vulnerable neurons (*Phactr1*, *Ptprd*, and *Celf1*). This demonstrated inter-dataset consistencies in the vulnerability-associated expression of these transcripts (Fig. 8i–k).

## Discussion

Advances in single-cell technologies are providing insights into the selective neuronal vulnerability that appears to span neurodegenerative disorders, which may lead to the development of therapies that slow the progression of a wider range of diseases. Here, we constructed a single-cell RNAseq atlas of live neurons isolated from prion-infected mice and used this unique resource to identify and characterize degenerating neuronal subsets. By leveraging the depletion of synaptic gene expression as a metric of neuronal damage in prion and Alzheimer's disease, we correlated baseline gene expression with neuronal vulnerability. Key findings included the robust upregulation of *Rbm3*, a neuroprotective stress-response gene[41,42], across several distinct neuronal subsets upon prion infection. Furthermore, *Rbm3* was highly expressed by hippocampal glutamatergic neurons that appear particularly prion-resistant. We identified additional

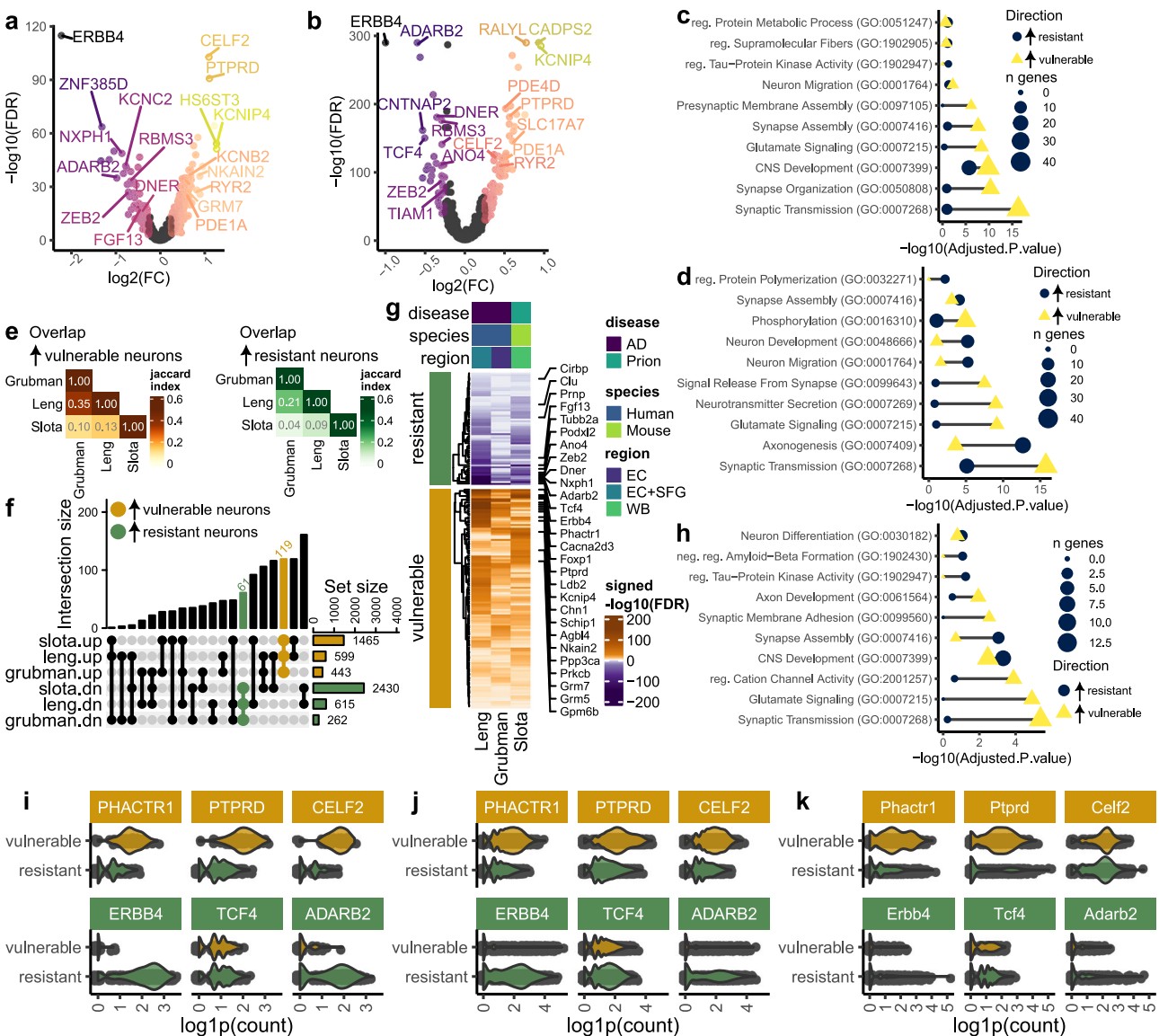

**Fig. 8 | Molecular correlates of neuronal vulnerability partially overlap between prion and Alzheimer's disease.** We identified baseline transcriptional differences that were correlated with neuronal vulnerability in Alzheimer's disease by analyzing two snRNAseq datasets previously published by Grubman et al. and Leng et al. (Supplementary Fig. 10). Volcano plots show vulnerability-correlated transcripts identified via MAST's two-part hurdle model in the (**a**) Grubman et al. and (**b**) Leng et al. datasets. Enrichr's Fisher exact test was used to compute gene ontologies enriched with transcripts that were significantly associated with vulnerable and resistant neurons in the (**c**) Grubman et al. and (**d**) Leng et al. datasets. Vulnerability-correlated transcripts (identified via MAST's likelihood-ratio test)

were next compared between the Grubman and Leng Alzheimer's Disease datasets with our prion (Slota) dataset. **e** Heatmap and (**f**) upset plot show the overlap of vulnerability-correlated transcripts identified in prion disease and Alzheimer's disease. **g** Heatmap shows a subset of vulnerability-correlated transcripts that were consistently increased in resistant and vulnerable neurons across all three datasets. **h**) Enrichment of gene ontologies was compared between shared markers of resistant and vulnerable neurons using Enrichr's Fisher exact test. Violin plots show expression of the top 3 transcripts associated with vulnerable and resistant neurons (identified via MAST's likelihood-ratio test) in the (**i**) Grubman et al., (**j**) Leng et al., and (**k**) Slota et al. (this study) datasets.

vulnerability-correlated transcripts that have previously been linked to prion disease, such as *Grm5*[46], *Grm8*[46], *Bmerb1*[47], *Macrod2*[46], and *Stmn2*[48]. By associating these genes with neuronal vulnerability, our findings provide context to understand their role in neurodegeneration and their potential as targets for pharmacological manipulation. Moreover, the overlap of vulnerability markers between prion and Alzheimer's disease points to the possibility of targeting common disease mechanisms and raises questions about the influence of gene expression on neuronal susceptibility to multiple proteopathic seeds.

A key goal in neurodegenerative disease research is to identify specific neuronal subpopulations that degenerate. Early studies of prion disease used histological observation of brain tissues to

demonstrate the selective vulnerability of *Pvalb*[+] GABAergic neurons[6–9]. Our study revealed a more complete picture of prion-vulnerable neurons, encompassing classes of glutamatergic, GABAergic, and medium spiny neurons. The glutamatergic neuronal vulnerability was expected since signaling through glutamate receptors might contribute to prion toxicity[45]. More compelling were medium spiny neurons, which could serve as a focal neuronal subpopulation for assessing the effect of putative therapeutics on neurotoxicity since they were readily verified as vulnerable via RNA FISH. Interestingly, the well-established vulnerability of medium spiny neurons in Huntington's disease[51] further illustrates potential commonalities across neurodegenerative disorders.

We demonstrated that individual degenerating neurons stop expressing synapse-related genes in prion disease, which could play a contributing role in synaptic toxicity[20–22,52,53]. Our group[31] and others[54] have previously demonstrated temporal disruptions in synapse-related gene expression beginning at early disease stages. Prion pathogenesis culminates in physical disruptions to synaptic membranes, perhaps via oxidative stress[55], phagocytosis[56,57], or direct membrane interactions with PrPSc[58–60]. Thus, early synaptic changes might indicate remodeling to avoid stress and damage. This hypothesis should be evaluated by performing scRNAseq at earlier time points, despite the challenge of isolating sufficient cells to analyze the full range of neuronal transcriptional states. At late disease stages, the depletion of synapse transcripts most likely reflects a downstream response to synaptic dysfunction. Synaptic membrane damage might cause transcripts to leak out into the extracellular milieu to be degraded, or lead to long-term depression and transcriptional down-regulation of synaptic genes.

While the molecular mechanisms that link prion accumulation with synaptic damage remain unclear, our findings establish the pathway-level decrease of synapse-related gene expression as a useful metric for identifying and ranking prion-vulnerable neuronal subsets. Conclusions on neuronal vulnerability are often based on cell composition changes. Since synaptic damage can be detected early on[20–22,52,53], our approach that uses synaptic gene depletion can potentially identify a broad spectrum of damage-vulnerable neurons. This could be especially advantageous in mouse models of neurodegeneration because mice are sacrificed at relatively early disease stages when neuronal loss is minimal.

By associating gene expression with neuronal vulnerability, our study provides molecular insights into neurodegeneration. Most striking was the prion-associated upregulation of the cold-shock-response gene *Rbm3* across diverse neuronal subsets, which also exhibited higher baseline expression by prion-resistant neurons. Enhancing *Rbm3* expression in prion-infected mice is neuroprotective[41,42], painting *Rbm3* as a target for pharmacological manipulation. The cold-induced upregulation of *Rbm3* has been shown to be mediated by TrkB signaling[61] and is thought to enhance structural plasticity and promote synapse regeneration[41]. We did not detect neuronal *RBM3* expression in the Leng or Grubman Alzheimer's disease datasets, possibly because it is more challenging to detect low-abundance transcripts with single-nucleus RNA sequencing. Thus, we were unable to conclude whether *Rbm3's* association with neuronal vulnerability is common to prion and Alzheimer's disease. We also validated the prion-induced downregulation of the immediate early gene *Egr1* and the small GTPase *Rab6b* in cortical glutamatergic and striatal medium spiny neurons, respectively. As both *Egr1*[62] and *Rab6b*[63,64] are linked to neuronal function, their prion-elicited downregulation is likely part of the overall depletion of genes broadly related to synaptic function and neuronal identity that was seen across various neuronal subsets.

PrPSc undeniably precipitates prion toxicity, and PrPC might transduce neurotoxic signaling elicited by PrPSc, amyloid-β, and α-synuclein[65]. On the other hand, some evidence suggests that purified prions are not directly neurotoxic[66], and immortalized cell cultures appear healthy when actively replicating high levels of PrPSc[67]. In either case, the well-defined relationship between *Prnp* expression levels and PrPSc replication rates[68,69] lends to the intuitive assumption that *Prnp* transcript abundance might influence toxicity. Surprisingly, this postulate is challenged by our finding that *Prnp* was more abundant in neuronal sub-clusters classified as resistant to prion and Alzheimer's disease. As *Prnp* is expressed ubiquitously, it is unclear whether the naturally occurring *Prnp* expression differences across cell types are substantial enough to modify prion toxicity. Toxicity could also be modified by post-translational modifications of PrPC, like glycosylation and GPI-anchoring, which are not captured by transcriptomics.

In conclusion, our study demonstrates the power of single-cell transcriptomics to resolve and characterize prion-vulnerable neuronal subsets. Applying this technology to study prion disease is particularly challenging due to the constraints of working under biocontainment conditions and the considerable difficulty of isolating live neurons. Findings can be influenced by artefactual gene expression changes during tissue dissociation, and particularly sensitive neuronal subsets may be lost during ex vivo cell isolation. In addition, we expect many disease-associated neuronal changes to be localized to synapses and neurites, which may not be reflected by transcription at the cellular level. Sub-cellular alterations await confirmation by high-resolution microscopy. We also note that the number of neurons analyzed (~10,000) was comparatively modest. Generating larger datasets, potentially targeting additional cell types and brain regions at multiple time points during prion infection, will be valuable. As this study employed only one strain of mouse-adapted scrapie, analyzing and comparing neuronal vulnerability between different prion strains would strengthen our findings. In addition, translating these findings to human disease must be a future aim. Ultimately, bringing together multiple datasets, including the single nucleus and spatial transcriptomics, comparisons with other neurodegenerative disorders, and investigations beyond transcription will provide a comprehensive view of neuronal vulnerability and pinpoint new drug targets.

## Methods

### Mice

The Animal Care Committee of the Canadian Science Center for Human and Animal Health approved all animal experiments and procedures under the Animal Use Document (AUD) # H-20-024. Female CD1 mice (*mus musculus*) at 7 weeks of age (originating from the University of Manitoba's animal colony) were intraperitoneally inoculated with 200 μL of 2% brain homogenate from either Rocky Mountain Laboratory Scrapie-infected (RML) or non-infected (Mock) mice. Mice were housed with a light/dark cycle of 12/12 h at 21 °C and relative humidity of 50%. As described previously[30–32,39], the mice were monitored for clinical signs that include dull ruffled coat, pinched abdomen, weight loss of 20% or more, and ataxia. Upon reaching the clinical endpoint, mice were deeply anesthetized using isoflurane gas and sacrificed via transcardial perfusion with PBS. The five RML-infected mice used for single-cell RNA sequencing reached the clinical endpoint at 164, 172, 178, 179, and 182 days post-infection (dpi), while the four Mock-infected mice were sacrificed at 148, 168, 185, and 189 dpi. Thus, the mice were of an average age of ~210 days old when brains were collected.

A separate set of three RML- and three mock-infected mice collected at clinical endpoint were used for RNA fluorescence in situ hybridization. These mice were deeply anesthetized using isoflurane gas and then transcardially perfused with 10% neutral buffered formalin. Brains were collected, stored in 10% formalin, and then paraffin embedded.

### Preparation of neuron-enriched live single-cell suspensions

Infectious tissues were handled within the CL2 + prion containment suite at the Canadian Science Center for Human and Animal Health. Immediately after mice were PBS-perfused, brains were removed and immersed in ice-cold loading buffer (Ca²⁺ and Mg²⁺-free PBS + 0.04% BSA + 0.6% glucose + 1 mM Kynurenic acid). The olfactory bulb was left behind when brains were removed, and we manually removed the cerebellum/hindbrain, leaving the forebrain and midbrain that were carried forward. The forebrain and midbrain tissues were collected in a glass petri dish, minced with a scalpel, and dissociated in 5 mL Hibernate e minus calcium with 20 units/mL papain and 0.005% DNase I for 40 min at 37 °C. To promote dissociation, tissues were gently triturated through a wide-bore (~5 mm) serum-coated plastic Pasteur pipette every 10 min. After the 40-minute incubation, the tissue was

triturated 10 times with a serum-coated standard p1000 pipette tip, and single-cell suspension was diluted with 35 mL of loading buffer. Large debris was initially removed by passing the single-cell suspension through a 70 µm MACS strainer (Miltenyi), after which cells were pelleted by centrifuging at $300 \times g$ for 5 min at 4 °C. Debris was removed from the cell pellet using debris removal solution (Miltenyi) as described previously[39]. Neurons were enriched from the debris-liberated cell suspension using the mouse neuron isolation kit (Miltenyi) according to the manufacturer's instructions. Briefly, the cell pellet was re-suspended in 320 µL loading buffer, mixed with 80 µL adult non-neuronal cell biotin-antibody cocktail, incubated on ice for 5 min, diluted with 4 mL loading buffer and centrifuged at $300 \times g$ for 10 min at 4 °C. The cell pellet was then re-suspended in 320 µL loading buffer, mixed with 80 µL anti-biotin microbeads, incubated on ice for 10 min, and then diluted with 1600 µL loading buffer. The labeled cells were then passes sequentially through two LS columns in the magnetic field of a MACS separator, which were sequentially washed with a total of 4 mL loading buffer. The final cell suspension was passed through a 70 µm MACS strainer (Miltenyi), centrifuged at $300 \times g$ for 10 min at 4 °C, and the cells were re-suspended in 50 µL of loading buffer and counted on a hemocytometer via trypan blue staining.

## Single-cell RNA sequencing

Live single-cell suspensions were processed into sequencing libraries in the Chromium Next GEM Single Cell 3′ Reagent Kit v3.1 (Dual Index) (10 × Genomics) as described previously[39]. Briefly, reaction mixtures (containing cells + cDNA synthesis reagents) and gel beads were loaded onto Chromium Next GEM Chip G (10 × genomics), partitioned on a Chromium Controller, and the resulting GEMs were further processed into sequencing libraries according to the manufacturer's protocol. Unique indices (Dual Index Plate TT Set A) were assigned to each library during the final PCR amplification step (15 cycles). Bioanalyzer High Sensitivity DNA kits (Agilent) were used to assess cDNA and sequencing library quality. Libraries were sequenced to a targeted depth of 60,000 reads per cell on Illumina NextSeq 2000 P3 flow cells (100 cycles) using the standard read configuration for 10x genomics 3′ gene expression libraries (Read 1–28 cycles; i7 Index−10 cycles; i5 Index−10 cycles; Read 2–90 cycles). Up to 10,000 cells were sequenced per run, and 10 and 11 runs were performed on single-cell suspensions made from five prion and four mock-infected mice, respectively (up to four runs were performed per mouse). Sequencing metrics for each library are provided in Supplementary Data 1. Across these 11 libraries, we sequenced a total of 134,205 cells at an average depth of 62,007 reads/cells.

## scRNAseq data preprocessing and normalization

Single-cell RNA sequencing data was analyzed using our previously described bioinformatics pipeline[39]. Data was first pre-processed and quality-controlled individually for each library. Raw sequencing reads were pre-processed with cell ranger (v7.1.0) by de-multiplexing raw bcl files with mkfastq and then generating gene expression matrices with cell ranger 'count' (with mouse mm10 reference genome). Ambient RNA reads were then removed using soupX[70] (v1.6.2), low-quality cells were removed via filtering criteria (n genes > 1000, % mitochondrial genome reads < 20, % ribosomal protein reads > 1, percent hemoglobin reads < 20), and doublets were removed using DoubletFinder[71] (v2.0.3; default settings - pN = 0.25, pK = 0.09, and 7.6% expected doublets). Quality control metrics for each library are provided in Supplementary Data 1. After filtering, a total of 100,946 cells remained, with an average of 2187 genes detected per cell.

Seurat[72,73] (v4.2.5) was then used to normalize the cleaned datasets using SCTransform()[74] according to a Gamma-Poisson Generalized Linear Model, and the percentage of mitochondrial reads was regressed out. The full single-cell transcriptional atlas was prepared by integrating each individual dataset using reciprocal PCA analysis. We randomly chose three prion-infected (neuM08, neuM10, and neuM13) and three Mock-infected (neuM05, neuM17, and neuM21) libraries as references for finding integration anchors. Cells were clustered using Seurat's graph-based clustering and visualized using UMAP, supplying the first 50 principle components for dimensionality reduction. Globally distinguishing marker transcripts of each cell cluster were identified with Seurat's FindAllMarkers() function using the default Wilcoxon Rank Sum test and Bonferroni-adjusting $p$-values against all genes in the dataset (Supplementary Data 2). Cell clusters were then preliminarily classified by brain cell type using SCType[75] (v1.0) and by manually inspecting marker genes. For more precise cell type classification, we then mapped the transcriptomes to the Allen Brain Atlas's high-resolution reference mouse brain cell atlas with their MapMyCells tool[43]. We considered cell classification according to SCtype and MapMyCells, and the expression of marker genes when assigning a final cell type to each cluster (Supplementary Fig. 1).

## Integration of neuronal transcriptomes to produce a neuronal single-cell atlas of prion infection

To produce a single-cell atlas of high-quality neuronal transcriptomes, we first identified 8148 transcriptomes from the full dataset that were classified as neuronal according to MapMyCells and belonged to the following neuronal clusters: gaba.neu.18, gaba.neu.19, msn.neu.23, msn.neu.28, glut.neu.17, glut.neu.21, glut.neu.25, and glut.neu.32. Similarly, we identified high-quality neuronal transcriptomes in our previously published single-cell RNAseq dataset[39], which was made according to the same methodology applied to the same RML mouse model. This live single-cell atlas of cortical and hippocampal brain cells during prion infection (Slota et al. 2022 dataset) is available from the broad institute (https://singlecell.broadinstitute.org/single_cell/study/SCP1962/dysregulation-of-neuroprotective-astrocytes-a-spectrum-of-microglial-activation-states-and-altered-hippocampal-neurogenesis-are-revealed-by-single-cell-rna-sequencing-in-prion-disease#study-visualize). This dataset contains 1248 neuronal transcriptomes isolated from the cortex, and 1358 neuronal transcriptomes isolated from the hippocampus. Starting from the soupX-corrected raw RNA counts, the 8148 transcriptomes from the present dataset were combined with the 2606 transcriptomes from our previous dataset[39] to produce an atlas of 10,754 neuronal transcriptomes. The neuronal atlas was then normalized with Seurat's SCTransform()[74], using the 'v2 regularization' method[76], and then integrated and re-clustered according to Seurat's CCA, RPCA, and harmony methods. The harmony integrated clustering was carried forward for subsequent analysis (Supplementary Fig. 2). Graph-based clustering (resolution = 0.8) and dimensionality reduction were then applied to the resulting integrated atlas, classifying 10,754 glutamatergic, GABAergic, and medium spiny neurons into 42 sub-clusters.

## Cell composition analysis

Cell cluster relative frequencies were compared between prion- and Mock-infected mice using the Bayesian model implemented by scCODA[77] (v0.1.9, Python = v3.11). Briefly, scCODA accounts for the compositionality of cell frequency data by fitting it into a Dirichlet-Multinomial model with a log-link function, modeling compositional changes in relation to a reference cell population. We utilized the "no-u-turns" sampling method for Markov-chain Monte Carlo (MCMC) based parameter inference. We employed scCODA's automated reference selection method. Results from the scCODA cell composition analysis (and the reference cell clusters chosen) are summarized in Supplementary Data 3.

## Differential expression analysis

To identify prion-altered and vulnerability-correlated transcripts, we performed differential expression analysis with MAST's two-part hurdle model[78] (v1.26.0), which models both the cellular detection rate and the mean of positive expression. SoupX-corrected raw UMI counts

were supplied for model construction with MAST. The number of genes detected per cell, the percentage of mitochondrial genome reads, and prion disease status were fitted to the model as covariates, with sample ID being fitted as a random effect to overcome pseudo-replication bias[79]. To minimize artefactual changes in neuronal gene expression and speed up computation time, we removed ubiquitously expressed transcripts that we have previously found to be highly variable, including *Malat1* and short transcripts encoding mitochondrial and ribosomal subunits[30,39]. We then computed differential gene expression between prion and mock-infected mice using the likelihood ratio test, adjusting *p*-values via the Benjamini-Hochberg method. Differentially expressed transcripts were defined based on FDR-corrected *p*-value < 0.05, and are provided in Supplementary Data 4.

### Functional enrichment and Gene set enrichment analysis (GSEA)

Enrichr[80,81] (v1.1.0) was used for functional enrichment analysis of provided gene lists against the GO-biological process database. Synapse-localized prion-altered genes were identified using SynGO[82] (v1.2).

We also performed GSEA on differential expression results for each neuronal sub-cluster with the fgsea R package (v1.26.0), supplying signed −log10(DE FDR *p*-values) for pre-ranked GSEA against the GO biological processes.

Escape[83] (v1.1.0) was used for single-cell gene set enrichment analysis (scGSEA) of the sub-clustered neuron dataset against the following gene sets: "GOCC_SYNAPSE", "GOBP_REGULATION_OF_SYNAPTIC_PLASTICITY", "GOBP_NEUROTRANSMITTER_SECRETION", and "GOBP_AXON_DEVELOPMENT". Gene set enrichment scores were compared between prion- and mock-infected cells using Escape's getSignificance() function with default settings.

### Gene set enrichment analysis of previously published bulk and single-cell RNAseq datasets

The following bulk RNAseq datasets of prion infection were downloaded from NCBI GEO: GSE144738[24], GSE149805[33], GSE201249[30], and GSE90977[44]. Raw fastq files were downloaded for each dataset and were subsequently pre-processed using our previously described bioinformatics pipeline[30]. Briefly, reads were cleaned using Trimmomatic, rRNA reads were removed using Bowtie2, reads were mapped to the mouse Grcm38 reference genome using HISAT2, and known transcript reads were counted with FeatureCounts. DESeq2 was used to normalize the resulting gene expression matrices and perform differential expression analysis. For each comparison, T-statistics of genes with baseMean > 25 and |log2FoldChange| > 0.1 were supplied for pre-ranked GSEA against synaptic gene sets using fgsea.

### Comparison of differential expression results

We compared the overlap of differentially expressed genes between each neuronal sub-cluster by calculating the jaccard index (intersection/union) of all positively and negatively prion-associated genes with corrected *p*-value < 0.1. As an additional means to compare prion-altered transcriptional responses between neuronal sub-clusters, we computed signed −log10(adjusted *p*-values) from the MAST differential expression results and correlated them using R's (v4.3.1 s) corr.test() function. We only included genes with FDR < 1 for correlation analysis and only reported a correlation between gene lists with > 100 matching transcripts after filtering.

### Prediction of vulnerable neuron sub-clusters and identification of vulnerability markers

Each neuronal sub-cluster was assigned a vulnerability score based on synaptic gene depletion. Vulnerability scores were calculated as the −signed(−log10(GSEA FDR)) for the "GOCC Synapse" gene set, taken from the GSEA results computed from the MAST differential

expression results. Thus, high vulnerability scores demarked the neuronal subsets that elicited a more pronounced prion-associated depletion in synaptic gene expression. For visualization purposes, neuronal sub-clusters with a vulnerability score >1 were classified as vulnerable, corresponding to a decreased GOCC Synapse pathway enrichment with FDR = 0.1.

Vulnerability markers were then identified through differential expression analysis against vulnerability scores, using neurons isolated from mock-infected mice only. MAST's two-part hurdle model was used for this differential expression analysis by assigning vulnerability score, the number of genes detected per cell, and the percentage of mitochondrial genome reads as covariates in the model, and fitting sample ID as a random effect[79]. Vulnerability-correlated transcripts were identified by using vulnerability score as the variable for MAST's likelihood ratio test. As prion-infected mice were not used for this analysis, these vulnerability markers reflect underlying gene expression differences between neuronal sub-clusters with high versus low vulnerability scores. Vulnerability-correlated transcripts are provided in Supplementary Data 5.

### Analysis of neuronal vulnerability in previously Alzheimer's disease datasets

For comparison with prion disease, we also identified vulnerability-correlated transcripts in previously published Alzheimer's disease datasets. Single nucleus RNAseq datasets of Alzheimer's disease were downloaded from GEO: GSE138852 (Grubman et al. dataset)[50] and GSE147528 (Leng et al. dataset)[36]. Starting from raw gene expression matrices, each dataset was normalized and integrated with Seurat as described above, and the neuronal sub-clusters from the original study were retained. We next performed differential expression analysis using MAST's two-part hurdle model, supplying raw UMI counts for model construction with MAST. The number of genes detected per cell, the percentage of mitochondrial genome reads, and Alzheimer's disease status (Grubman dataset) or Braak score (Leng dataset) were fitted to the model as covariates, with sample ID being fitted as a random effect. Using MAST's likelihood ratio test, we identified lists of transcripts that were associated with Alzheimer's disease (Grubman dataset) or Braak score (Leng dataset) for each neuronal sub-cluster. We next performed pre-ranked GSEA on the lists of differentially expressed transcripts, using the signed −log10(DE FDR) as the ranking metric. Each neuronal sub-cluster was then assigned a vulnerability score based on AD-associated negative enrichment for the GOCC_SYNAPSE gene sets, calculated as vulnerability score = − signed(-log$_{10}$(GSEA FDR)). Finally, vulnerability-correlated transcripts were identified through differential expression analysis with MAST as described above, using vulnerability score as the variable for MAST's likelihood ratio test. To prevent AD-associated expression changes from influencing the vulnerability correlates, we first subset the Grubman dataset to only include neurons from control cases, while the Leng dataset was a subset to only include AD cases with a Braak score of 0. For comparison of these AD vulnerability markers with prion disease, gene symbols in the human AD datasets (Grubman et al. and Leng et al. datasets) were first converted to the equivalent mouse gene symbol.

### RNA fluorescence in situ hybridization (RNA FISH)

Formaldehyde-fixed paraffin-embedded (FFPE) brains were sectioned coronally into 5 μm slices. Coronal sections were made at the levels of the (a) striatum, (b) hippocampus, and (c) hindbrain/cerebellum. One representative section of each of these three areas (a-c) was placed per microscope slide and further processed together. The sections were stained with RNA FISH probes using the Multiplex Fluorescent Reagent Kit v2 (RNAscope™, Cat. No. 323100) according to the manufacturer's instructions. Briefly, FFPE sections were first deparaffinized via two washes in Xylene, followed by two washes in 100% ethanol. Sections

were then pre-treated with hydrogen peroxide for 10 min, incubated in target retrieval reagent for 30 min at ~ 99 °C in a steamer, and then treated with RNAscope™ protease-plus for 30 min at 40 °C in a HybEZ oven. Sections were then hybridized with RNAscope™ probes for 2 hrs, followed by 3 cycles of probe amplification and HRP signal development with the corresponding fluorescent dye: Opal™ 520 (Akoya Biosciences, Cat. No. FP1487001KT), Opal™ 570 (Akoya Biosciences, Cat. No. FP1488001KT), and Opal™ 650 (Akoya Biosciences, Cat. No. FP1496001KT). Finally, sections were mounted with Prolong Glass with NucBlue (ThermoFisher Cat. No. P36983) and cured for 24 h at room temperature.

Three RNAscope™ probe panels were used to assess cell composition. These panels targeted glutamatergic neurons (RNAscope™ Probe-Mm-Slc17a7-O2 Cat. No. 501101, RNAscope™ Probe-Mm-Slc17a6-E1-E3-C2 Cat. No. 456751-C2, and RNAscope™ Probe-Mm-Grin2b-C3 Cat. No. 417391-C3), GABAergic neurons (RNAscope™ Probe-Mm-Sst Cat. No. 404631, RNAscope™ Probe-Mm-Gad1-C2 Cat. No. 400951-C2, and RNAscope™ Probe-Mm-Pvalb-C3 Cat. No. 421931-C3), and medium spiny neurons (RNAscope™ Probe-Mm-Gad1-C2 Cat. No. 400951-C2, and RNAscope™ Probe-Mm-Drd1-C3 Cat. No. 461901-C3). For cell composition analysis, slides were imaged using a Zeiss Axioscan 7 Microscope Slide Scanner. Slide scanning microscopy produced tile images, which were shading-corrected using Zeiss Zen Blue software (v3.8) with the "shading reference from tile image", and the "shading correction" methods. Regions of interest (ROIs) were then selected from the shading-corrected images, targeting brain regions relevant to each panel of RNAscope™ probes. Glutamatergic neuron panel: cortex, hippocampus (CA1), midbrain and thalamus. GABAergic neuron panel: cortex, cerebellum, hippocampus (dentate gyrus), midbrain, and thalamus (reticular nucleus). Medium spiny neuron panel: striatum. The number of ROI images and cells analyzed in each panel is provided in Supplementary Data 6.

Six RNAscope™ probe panels were used to measure gene expression. To assess Glutamatergic and GABAergic neurons, RNAscope™ Probe-Mm-Slc17a7-O2 Cat. No. 501101 and RNAscope™ Probe-Mm-Gad1-C3 Cat. No. 400951-C3 were combined with RNAscope™ Probe- Mm-Ttr-C2 Cat No. 424171-C2, RNAscope™ Probe- Mm-Rbm3-C2 Cat No. 519161-C2, or RNAscope™ Probe- Mm-Egr1-C2 Cat No. 423371-C2. To assess medium spiny neurons, RNAscope™ Probe- Mm-Drd2 Cat No. 406501 and RNAscope™ Probe- Mm-Drd1-C3 Cat No. 461901-C3 were combined with RNAscope™ Probe- Mm-Ttr-C2 Cat No. 424171-C2, RNAscope™ Probe- Mm-Rbm3-C2 Cat No. 519161-C2, or RNAscope™ Probe- Mm-Rab6b-C2 Cat No. 455291-C2. For gene expression analysis, these slides were images on a Zeiss LSM980 confocal microscope. High-resolution ROI images were collected using LSM980's airyscan mode with a 20x objective. For Glutamatergic/GABAergic neurons, ROIs were targeting the cortex, CA1, and dentate gyrus regions. For medium spiny neurons, ROI's were targeted in the striatum.

ROI images were analyzed as follows. Background signal was manually removed for each ROI, which were then exported as tiff images for subsequent analysis with Cellprofiler (v4.2.5). A custom cellprofiler pipeline was used to count individual nuclei in each ROI image and count RNA puncta for each probe. Briefly, nuclei were segmented via the "identify primary objects" module using the adaptive thresholding strategy and the minimum cross entropy threshold method. Cells were then defined using the "identify secondary objects module", based on a distance of 20 pixels relative to each nuclei. For each channel, RNA puncti were identified with the "identify primary objects" module and related to each cell object. Signal intensity of each channel was then measured in each cell using the "measure object intensity" module, and morphological features of each cell/nuclei were measured using the "measure object size shape" module. Data from the cell profiler was exported in CSV format for subsequent statistical analysis and plotting using custom R scripts. RNA expression was calculated as the density of puncti per mm$^2$ within each cell. Nuclei were defined as positive for the expression of each transcript based on log$_2$(puncti density) > 5. This was used to calculate the total proportion of nuclei positive for each transcript and to assign nuclei to cell classes. Glutamatergic neurons were defined by log$_2$(Slc17a7 puncti density) > 5, GABAergic neurons were defined by log$_2$(Gad1 puncti density) > 5, and medium spiny neurons were defined by log$_2$(Drd1/2 puncti density) > 5.

The Shapiro-Wilk normality test was used to determine whether data was normally distributed. Cell proportions from RNA FISH data were compared between RML and mock-infected mice using the Wilcoxon rank sum test with continuity correction. For comparing gene expression, a pseudobulk analysis was achieved by taking the average log$_2$(puncti density) of all nuclei assigned to a given cell type per individual mouse ($n = 3$). The per-mouse averages of log$_2$(puncta density) were normally distributed and compared between prion- and mock-infected mice using the student's $t$ test. Single-cell level RNA expression measurements are provided in Supplementary Fig. 9g.

### Data visualization

Seurat was used to prepare UMAP plots and violin plots. Upset plots and hierarchical clustered heatmaps were prepared using ComplexHeatmap[84] (v2.18.0). The default Pearson distance method was used for the hierarchical clustering of heatmaps. Other plots were prepared using ggplot2[85] (v3.4.2). Log1p(SCT-corrected UMI counts)[74,76] were supplied for visualizing single-cell level gene expression in heatmaps and violin plots.

### Reporting summary

Further information on research design is available in the Nature Portfolio Reporting Summary linked to this article.

## Data availability

All new single-cell RNA sequencing data generated by this study are available on the Broad Institutes single-cell portal as study # SCP2450. In addition, raw gene expression matrices for each individual library are available from Gene Expression Omnibus under accession number GSE249382. The RNAscope data generated by this study are provided in Source Data 1. The previously published single-cell atlas of live cortical and hippocampal brain cells during prion infection is available from the broad institute as study # SCP1962. Single nucleus RNAseq datasets of Alzheimer's disease are available from NCBI GEO: GSE138852 and GSE147528. The following previously published bulk RNAseq datasets of prion infection are available from NCBI GEO: GSE144738, GSE149805, GSE201249, and GSE90977. Source data are provided in this paper.

## Code availability

Custom R scripts and Python code are available at the following GitHub repository: https://github.com/jslota/scRNAseq_prion_neuronal_vulnerability.

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

## Acknowledgements

The Public Health Agency of Canada (PHAC) funded this work. We thank the staff of the NML veterinary technical service for assistance with animal manipulations and the staff of the NML sequencing core for assistance with Illumina sequencing runs.

## Author contributions

S.B. and J.S. conceived the project. J.S. performed single-cell RNA sequencing and data analysis. L.L. performed RNA-FISH. K.F. collected and processed tissues for RNA-FISH. B.S. provided input, expertize, and assistance with RNA-FISH and microscopy. J.S. acquired and analyzed microscopy images for RNA-FISH. J.S. prepared figures and wrote the manuscript draft. S.B. supervised the project. All authors read, edited, and approved the final manuscript.

## Competing interests

The authors declare no competing interests.
