## [Transparent Peer Review file · Nature Communications]

Single-cell transcriptomics unveils molecular signatures of neuronal vulnerability in a mouse model of prion disease that overlap with Alzheimer's disease

Corresponding Author: Dr Stephanie Booth

Version 0:

Reviewer comments:

Reviewer #1

(Remarks to the Author)

Slota et al.'s revision included important improvements to their manuscript. Their data, analysis, and presentation is certainly more accessible. They now present an interesting study of good quality and provide valuable insights.

To clarify from my previous response; I agree in the authors' choice to perform live neuron dissociation rather than nuclei isolation, and speculate that including the cytoplasmic transcripts was even key in the context of this study. I also fully understand that isolating viable neurons is tricky from these mice (although I'm convinced future studies will greatly benefit from the suggested improvements). It is unfortunate that relatively few neurons were captured, and that - given the ambitious sampling - the diversity of populations is very limited. Nevertheless, the dataset is unparalleled and valuable, and the analysis is valid as it is consistently matched.

The authors certainly succeeded to strengthen their neuronal data by adding published cortex and hippocampus data and performing new clustering. A key improvement was the mapping and naming of these clusters with the help of an existing, annotated atlas from Allen. This also further clarifies which neuronal populations were effectively included. Could the authors clarify whether, or in what way, these changes led to rather different key findings between the previous manuscript, and revision? I missed a summary clarifying these updates in the response letter. For example, - Where did the "Slc7a11" neuronal clusters from the initial manuscript map? These clusters were ascribed a central role in prion vulnerability. - S100b, Il33, Trdn, and Cybrd1 were named as differentially expressed in prion disease in accordance with the literature; these do not appear in the current version, while Rbm3, Ttr, IEGs and others had not been described in the previous analysis.

It was especially important that the authors now explicitly name, display and discuss individual genes altered in prion disease - rather than just general signatures or processes.

The newly added validation of some of these genes by Scope, and against published bulk datasets, in Figure 4 is overall convincing. Additional clarification of the statistics in Fig. 4g will be helpful (I understand from the methods validation came from 3 mice; how many sections? how many cells?)

Finally, I appreciate the authors' clarifications in the text. For example, the first paragraph now openly describes the disease model, sampling approach, how cell type abundances are out of balance, and how neurons were integrated with an existing neuronal dataset.

Also the Scope validation of MSN/GABA depletion is now better explained and justified. Changes in the method sections were important, this section is now much more clear.

(Remarks on code availability)

Reviewer #3

(Remarks to the Author)

Slota et al used single cell RNA Seq technology to analyse gene expression profiles of enriched cerebral neuron populations from CD1 mice infected with RML mouse-adapted scrapie prions, or mock infected control mice. They identified distinct gene expression responses in neuron subpopulations, and used level downregulation of synaptic gene expression to score neuronal damage and identified subsets of prion vulnerable neurons. Further, they compared gene expression profiles of prion-vulnerable and prion-resistant neurons and found that differences in baseline gene expression could influence neuronal vulnerability, highlighting increased expression of the neuroprotective cold shock protein Rbm3 in resistant neurons. Altered gene expression of selected candidates was validated using RNAScope analysis using brain sections of prion-infected and mock-infected mice. Moreover, comparison of their datasets to previously published transcriptomes from Alzheimer's disease models revealed that transcripts related to vulnerability overlap in Alzheimer's and prion disease models.

The manuscript has been peer-reviewed by two experts and this reviewer was asked to comment on the authors responses to previous concerns.

Reviewer 1:

(1) Data quality and comprehensiveness

The authors sequence >130k cells extracted from 'cerebral' brain tissue after depletion of non- neurons, and >100k cells meet relevant quality criteria. Out of these 100k cells, only a small minority is neuronal (Fig. 1a); and in fact, the remaining analysis is based on <9k cells with mature neuronal signatures. This ratio is far below what is expected – even considering that ex vivo stress during dissociation disproportionately affects neurons. I acknowledge that from this particular tissue it may be difficult to keep neurons viable, also considering the mice' likely advanced age at the time of sampling (although ages are not provided). Yet, even without neuronal enrichment, I would not expect more than 2x more non-neuronal than neuronal cells in the cerebrum, perhaps 3x in the striatum. The relative high number of endothelial cells is particularly unusual. In the same way, it is very unexpected that the authors would detect so many neurons with immature signatures. The immature neuron population looks like it is derived from the SVZ/RMS, but appears substantially overrepresented, again pointing towards unfavorable dissociation conditions for mature neurons, compared to the less vulnerable cell types.

The remaining dataset of ~9k neurons that Slota et al. analyze are consistently matched and compared between a relevant control and the disease state, certainly making those comparisons valid in principle. However, given the massive neuronal loss, it appears likely that many sensitive neurons and neuronal populations were selectively affected, questioning the comprehensiveness of the dataset.

Responses: Isolation of intact live neurons from prion infected mice was indeed challenging, and we suspect that endothelial cells and immature neurons were not effectively depleted. The bead based

neuronal enrichment kit from Miltenyi Biotech utilizes a proprietary mixture of antibodies that target non-neuronal cell surface markers. It seems likely that the antibody mixture was ineffective at removing endothelial cells and immature neurons in our case, which would explain the abundance of these cell types in our dataset.

We agree that the dataset lacked comprehensiveness, particularly in the case of thalamic/hypothalamic neurons that were absent from our dataset. Even the high resolution Allen Brain atlas reference atlas was supplemented with single nucleus RNAseq data for challenging- to-isolate neurons from the midbrain and hindbrain (Yao et al. 2023). This is a limitation that we were unfortunately not able to get around. In our case, cell isolation was performed on prion infected brain tissues within a containment suite with limited available equipment. Crucially, neither an ultracentrifuge nor a fluorescent cell sorter were available, which are necessary for single-nucleus based approaches. Thus, our only option was to isolate live neurons from prion- infected mice at an advanced age (mean ~210 days). We believe however, that this has resulted in a unique dataset with higher quality gene expression measurements compared to the single- nucleus datasets that are frequently used to study neuronal vulnerability in other neurodegenerative disorders (e.g. ref's 36-38 in the revised manuscript). The quality of the gene expression measurements and differential expression analyses is backed by our RNA-scope based validation of prion-altered transcripts that we have now included in the manuscript (Figure 4). We are not aware of any other studies that have used live single cell RNA sequencing in the study of neuronal toxicity or vulnerability, making our dataset a unique resource for the neuroscience community.

R3: This is a major concern that has also been raised by reviewer 2, and a challenge which the authors were not able fully address because it would essentially mean to repeat the entire experiment. However, despite the technical challenges the authors provide a highly relevant and unique dataset that enabled them to identify vulnerable populations of neurons, and baseline gene expression profiles that are related to resistance to prion-induced damage. Due to the long incubation time of prion disease in mice, naturally the disease models are at an advanced/old age at the time they reach clinical disease, which poses an additional challenge for isolating live neurons. However, the experiment is well controlled by using age matched mock-infected control mice for isolation of neurons according to the same protocol used for prion-infected mice to rule out gene expression changes induced by the relatively long dissociation time.

(2) Cell type analysis

Slota et al. find 21 neuronal cell types in their analysis, and annotate them as GABA, GLUT or MSN, and 3-4 marker genes. However, there is no attempt to map these cell types with respect to existing atlases (Allen Mouse Brain Yao et al. 2023,

Linnarsson datasets, etc.). For example, there was no mention of what the Slc7a11-neurons that are analyzed in more detail might be – and Slc7a11 is a marker much more expressed in the vasculature (VLMC, Pericytes, perhaps astrocytes). Further, in scoring the cell types according to vulnerability, the authors later state that there is no relation between vulnerability and neuronal subtype (and I believe they refer to classes, GABA, GLUT or MSN). For lack of a comprehensive attempt to understand which subtypes they describe this statement becomes less convincing.

Response: Thank you for this suggestion. We have now annotated cell types based on mapping cells to the Allen Brain Atlas's high-resolution mouse brain cell atlas (Yao et al. 2023). The major neuronal classes captured by our dataset corresponded to cortical, hippocampal, and anterior olfactory nucleus glutamatergic neurons, nucleus accumbens GABAergic neurons, striatal medium spiny neurons, with a smaller number of Pvalb+, Sst+, Sncg+, and Lamp5+ GABAergic neurons.

R3: This response is satisfactory.

(3) Data presentation

The manuscript would greatly benefit from scaling down on presenting lots of 'compressed' data; i.e. gene modules, scores, processes, unannotated heatmaps. Instead, at least examples of more 'raw' data would help interpret the data in a more nuanced way, such as to be able to judge effect sizes, and significance. The authors don't show many examples of genes, and they are often hidden in very dense heatmaps. Instead, they could choose to show differential expression of genes in a specific cell type as volcano plot or similar. There is not much mention of specific genes until the Discussion, and the genes highlighted in the Discussion are not shown in the data (main figures or supplementary).

Response: Thank you for this insight. It can be challenging to choose appropriate data visualizations for a dataset of this complexity. In the course of revising this manuscript, we have extensively revised the figures to include more examples of raw data through volcano plots and violin plots etc.

R3: This reviewer is unable to assess how much data presentation has improved as they have not seen the original manuscript. Current data presentation is appropriate.

(4) Validation

Slota et al. performed RNA Scope to quantify canonical cell classes and their expression intensity. This validation feels almost unrelated to the scRNA-seq dataset, as the authors do not describe that the canonical cell types or markers are depleted outside the RNA Scope analysis. It would be more relevant to focus on a few specific cell types, e.g. the Slc7a11 population, validate where it is located, and how the expression of 1-2 highly DEGs is affected over the course of the disease.

Response: As mentioned in our response to your first comment, our live scRNAseq dataset is not well suited to detect changes in cell composition. Thus, we were not surprised that changes in cell type abundance were not reflected. However, in figure 6 we demonstrate that the depletion of medium spiny neurons in the RNA scope analysis is consistent with them being identified as especially vulnerable to prion infection in our scRNAseq analysis.

We have now conducted a second RNA scope analysis to validate the prion-altered expression of four promising genes – Ttr, Rbm3, Egr1, and Rab6b. These data are presented in figure 4, and we detected robust prion-alterations of Rbm3, Egr1, and Rab6b. We would not expect to see much in the way of temporal gene alterations over the course of the disease, since neuronal loss/dysfunction is usually only detected and the experimental endpoint in murine models of prion disease. Instead, we demonstrated that these genes are prion-altered only near the endpoint of prion disease using previously published bulk RNAseq datasets that longitudinally examined prion infection. These late changes in neuronal gene expression are not surprising since, due to animal welfare regulations, prion-infected mice are sacrificed at relatively early disease stages when neuronal loss is minimal.

R3: The authors have appropriately addressed the concerns.

As general good publication practice, I would expect better description of the methodology throughout the main text. For example:

- No age of disease induction (or sacrifice) is provided.

Response: We have included these details in the methods section. Mice were inoculated at 7-weeks old, and were sacrificed between ~150-190 days post inoculation, meaning they were an average age of ~210 days old at the time of sacrifice. The RML mouse model we employed is extensively used within the prion field, and is essentially a standard model. This is why only minimal details are included in the main text.

- There is no mention of the dissection strategy in the main text, in Methods the authors write 'cerebral' tissues were collected without detailing which precise structures were included in the data. (Here, it might also be useful to see histological comparisons of how the 'cerebral' brain tissue looks at clinical endpoint, vs. mock injected, justifying the dissection strategy.)

Response: We apologize for the confusion with respect to the brain tissues that were used. We essentially used the whole brain, but removed the cerebellum and hindbrain to reduce sample complexity, and because relatively few pathological changes are detected within the cerebellum in this RML mouse model. Furthermore, the olfactory bulb was left behind when brains were removed. Thus, the remaining brain tissue after dissection corresponded to the entire forebrain + midbrain. This mouse model has been extensively examined in previous histological studies. For a particularly comprehensive histological study, see: <https://doi.org/10.1177/0300985819861708>.

- Analysis methods are barely described in the text at all. Without going into details, I expect an outline / intuition of the authors' intent in any new analysis when following the main text. An extreme example of this are the different analyses compared in Figure 7c-d.

Response: The revised manuscript includes additional methodological details in the main text. This has made the manuscript lengthier, but we hope the robustness of the dataset and analysis are now easier to judge.

R3: The authors have added more details as requested.

Reviewer #2:

1. The cell isolation protocol is flawed. Cells were dissociated for 40 min at 37 °C which result in artifactual changes in gene expression patterns as reported by multiple publications (<https://www.ncbi.nlm.nih.gov/pmc/articles/PMC5665481/> ; <https://genomebiology.biomedcentral.com/articles/10.1186/s13059-019-1830-0>).

Response: We agree that the extended dissociation step was a limitation of our protocol for cell isolation. See our responses to reviewer # 1 for a more detailed explanation of why we chose to isolate live neurons from prion infected mice. However, cells were isolated from both prion- and mock- infected mice according to the same protocol, enabling comparisons between the two groups.

Furthermore, we have now validated several prion-altered transcripts using RNA scope (Figure 4). This confirms the robustness of the dataset and verifies that the observed gene expression changes are 'real', rather than artefactual.

R3: See response to reviewer 1 comment.

2. The authors used vulnerability score cutoff of 0.5 to determine whether a neuronal population is vulnerable or resistant for their dataset. They adjusted vulnerability score thresholds "to 1.14 and 0.3 for the Grubman et al. and Leng et al. datasets respectively." The purpose is "to ensure a comparable number of vulnerable and resistant neurons for analysis". All the analysis criteria were arbitrary without any scientific justification.

Response: For identification of vulnerability-correlated transcripts, we have now performed a differential expression analysis based on the quantitative vulnerability score value, instead of using an arbitrary classification of neurons as either 'resistant' or 'vulnerable' to group cells for differential expression testing. The classification as either 'resistant' or 'vulnerable' is now only used for visualization purposes, and we now have adjusted the vulnerability score cut-off to 1 because this is equivalent to a synapse-gene module depletion with FDR = 0.1. In other words, neurons are classified as vulnerable if they display a depletion of synapse-gene expression that reaches (or nearly reaches) statistical significance.

R3: The authors have adjusted their analysis and provide sufficient justification for their approach.

3. Fig. 1b: no distinguishing marker for epen.15, vlmc.26, npc.30, peri.31, or oligo.33 cell type is shown.

Response: To show expression of more marker genes, Figure 1b displays expression of numerous marker genes as a heatmap, and Supplementary Figure 1 demonstrates marker gene expression for each major brain cell type in a violin plot.

R3: Heat maps and additional violin plots as shown in main and supplementary figure are satisfactory.

4. Fig. 1: endothelial cells were divided into 10 clusters but no explanation of genes that distinguish different subpopulations.

Response: Endothelial cells are not the focus of our investigation, which is why we have not went to the extent of defining the different clusters in this manuscript. However, we have now provided genes that globally distinguish cell clusters in the full scRNAseq atlas as supplementary data file 2.

R3: The argument that endothelial cells are not the focus of their study is justified.

5. Mouse cortex neuronal cell types have been very well characterized with detailed layer- specific neuronal subtypes and marker genes. This study did not reference or identify any of these well-defined neuronal subtypes. It is unclear how credible their cell type classification is.

Response: We have improved our cell classification by mapping cells to the Allen brain atlas's high- resolution mouse brain cell atlas. See our response to reviewer #1 for further details.

R3: The improved cell classification addresses the reviewer's concern.

6. The authors identified prion-associated composition changes in glutamatergic and GABAergic neurons, but not in medium spiny neurons. However, in their analysis, msn.neu was assigned as the reference for the comparison. scCODA method used in the manuscript requires a reference to be able to identify compositional changes. scCODA can automatically select an appropriate cell type as the reference or uses a pre-specified reference cell type. It's unclear why the authors did not use the automatically reference selection method. Why msn.neu was used as the reference and how selection of reference affect the result and conclusion?

Thank you for this suggestion. We have now employed the automated reference selection method in scCODA. The specific cell clusters that were selected as reference in each cell composition analysis are provided in Supplementary data File 3.

R3: The authors followed the reviewer's suggestion.

7. "The five RML-infected mice used for single-cell RNA sequencing reached the clinical endpoint at 164, 172, 178, 179, and 182 days post infection (dpi), while the four Mock-infected mice were sacrificed at 148, 168, 185, and 189 dpi." Each mice has a different dpi, why? The Mock-infected mice were collected at a different time point than their treatment group animals. This seems to lost the meaning of serving as the "control" group.

Response: Prion-infected mice were sacrificed based on when they exhibit clinical signs of disease, rather than being sacrificed at a specific timepoint. This is why each mouse has a different dpi.

Furthermore, our cell isolation protocol could be applied only to one mouse per day. Therefore, the Mock treated mice were sacrificed on different days than the prion infected mice. We chose an age range of mock mice to match the prion infected mice, and this is standard practice for murine models of prion disease.

R3: This reviewer agrees with the authors. The age range of mock-infected mice is appropriate.

8. The authors performed pseudotime analysis on glut.neu.3, glut.neu.5, and glut.neu.10 subpopulations. Initially these clusters were described as distinct neuronal subpopulations. Later the authors suggested that they represented a glutamatergic neuronal subset undergoing disease- associated transcriptional changes. These are conflicting results without experimental evidence to support either. If they are distinct neuronal subpopulations it does not make sense to perform pseudotime analysis on these subpopulations. The pseudotime DEGs just represent genes differentially expressed between different subpopulations.

Response: Thank you for pointing this out, we apologize for this oversight. The pseudotime analysis has been removed from the revised manuscript because these cell clusters were annotated as 'mixed' when we applied the improved cell classification system.

R3: This is satisfactory.

9. Figure 5C: The authors quantified signal intensity and compared gene expression differences using individual neurons. It is unclear how many mice were sampled. The statistics should be calculated using individual mice instead of using individual cells.

Response: We used a separate set of three prion-infected and three mock-treated mice for RNAscope analysis. As mentioned, we have now included a second RNAscope dataset to validation prion- altered gene expression. We have also improved our image analysis pipeline by quantifying gene expression as puncti density per mm² rather than signal intensity. For these gene expression measurements, we took the per-mouse averages for statistics.

R3: The RNAscope analysis performed by the authors is highly relevant to confirm changes in gene expression, as it supports the argument of prion-induced gene expression changes rather than artificial changes induced during isolation and dissociation of neurons. The authors have further refined their analysis following the reviewer's suggestions.

10. The logic of the statement "Notably, prion-vulnerable neurons highly expressed the metabotropic glutamate receptors Grm5 and Grm8, known mediators of prion toxicity^{58,59}, while prion-resistant neurons expressed Gal3st160 and Stmn2, implicated as genetic risk loci for prion disease. These findings suggest that prion-vulnerable neurons tend to be more terminally differentiated and excitable, whereas resistant neurons are less mature and express genes associated with axon regeneration." is not clear.

Response: We apologize for the confusion. This section of the manuscript has been revised and now reads as follows: "We identified biological processes that were enriched with the list of transcripts differentially expressed by vulnerable and resistant neuronal subsets using Enrichr (Figure 7e). Prion- vulnerable neurons highly expressed transcripts linked to synaptic transmission, memory, and cellular adhesion, while resistant neurons abundantly expressed genes related to vesicle trafficking, negative regulation of supramolecular fiber assembly, and ubiquitin-mediated protein catabolism. Interestingly, several vulnerability-correlated transcripts have been previously linked with prion disease. For instance, the metabotropic glutamate receptors Grm5 and Grm8 mediate prion toxicity^{46,47}, while Bmerb148, MacroD247 and Stmn249 are implicated as putative genetic risk loci for prion disease. Several tubulin-encoding genes, including Tubb2a, Tubb2b, Tubb4a, Tubb4b, and Tubb5, were highly expressed by prion-resistant neurons. This is of note because in a previous study we found PrPC-overexpressing neuronal cultures, which were demarked by high tubulin expression, to completely resist prion infection⁵⁰. Moreover, the baseline expression of the neuroprotective stress-response gene Rbm3^{37,38} was higher in prion- resistant neurons (Supplementary File 4), and this was supported by RNA FISH that detected conspicuously low levels of Rbm3 expressed by selectively vulnerable medium spiny neurons (Figure 4g). The prion-associated upregulation of Rbm3 consistently seen across different neuronal subsets (e.g. Figure 3 and Figure 4) could imply a protective response. Thus, it is tempting to speculate that neuronal expression of specific genes could modulate vulnerability to prion-induced damage."

R3: This is a satisfactory response.

11. The authors found Galnt13 as the only vulnerability marker common to all four datasets. it is unclear whether this gene has anything to do with neuron vulnerabilities.

Response: Galnt13 catalyzes the O-linked glycosylation of mucins. We thought this was interesting because mucins have been shown to impair prion misfolding in cell-free conversion assays, presenting an analytical barrier for detecting prions in saliva, which contains mucins (PMID 29950332).

Galnt13 likely plays a different role in the brain, for instance, it may promote neurogenesis through glycosylation and stabilization of PDPN (PMID 12407114). Glycosylation is seemingly linked to prion replication, since heparin (PMID 24648544) and dextran sulfate (PMID 20957174) can serve as co-factors that modulate prion replication. Thus, it is plausible that the modulation of glycosylation by Galnt13 could influence prion replication, which could have a downstream effect on neuronal vulnerability. However, since we do not have strong data to support this hypothesis, the reference to Galnt13 has been removed from the text.

R3: This is indeed an interesting link that the authors might want to investigate in more detail in a follow up study. Removing the reference to Galnt13 is appropriate at this point.

Additional minor comments R3:

- Figure 1 legend: volcano plot is panel d (not c), hierarchical clustering is panel e (not d)
- in the discussion, the authors could provide further arguments for the use of live neurons rather than single nuclei RNASeq, as done in previous studies, that avoids some of the challenges faced with neuron enrichment. Did they identify transcripts that would have been lost if they had used snRNA-Seq?
- while such extensive scRNASeq studies cannot be performed in parallel with multiple prion strains, it is recommended to discuss the lack of validation with respect to different strains

(Remarks on code availability)

Reviewer #1 (Remarks to the Author):

Author response: We are most appreciative of the reviewer for their time in considering our manuscript and for providing such helpful and constructive comments.

Slota et al.'s revision included important improvements to their manuscript. Their data, analysis, and presentation is certainly more accessible. They now present an interesting study of good quality and provide valuable insights.

To clarify from my previous response; I agree in the authors' choice to perform live neuron dissociation rather than nuclei isolation, and speculate that including the cytoplasmic transcripts was even key in the context of this study. I also fully understand that isolating viable neurons is tricky from these mice (although I'm convinced future studies will greatly benefit from the suggested improvements). It is unfortunate that relatively few neurons were captured, and that - given the ambitious sampling - the diversity of populations is very limited. Nevertheless, the dataset is unparalleled and valuable, and the analysis is valid as it is consistently matched.

The authors certainly succeeded to strengthen their neuronal data by adding published cortex and hippocampus data and performing new clustering. A key improvement was the mapping and naming of these clusters with the help of an existing, annotated atlas from Allen. This also further clarifies which neuronal populations were effectively included.

Could the authors clarify whether, or in what way, these changes led to rather different key findings between the previous manuscript, and revision? I missed a summary clarifying these updates in the response letter. For example,

Author response: The broad findings and conclusions were largely similar in the revised manuscript. That is, the characterization of prion-altered gene expression in different neuronal populations, ranking neuronal vulnerability based on the depletion of neuronal gene expression, and the identification and comparison of vulnerability-correlated transcripts in prion and Alzheimer's disease. Many of the changes had to do with improving the robustness of the analysis methodology and changing the way the data and results were presented. As such, there were a few differences in terms of the specific cell clusters and genes that were focused on in the revised manuscript, which we discuss below in more detail.

- Where did the "Slc7a11" neuronal clusters from the initial manuscript map? These clusters were ascribed a central role in prion vulnerability.

Author response: The 'Slc7a11' neuronal clusters were assigned to cluster '1.Mixed.Glut' in the revised analysis. As mentioned in lines 130-135, we believe they represent damaged neurons that are sensitive to stress during cell isolation. They were more abundant in prion-infected mice (Figure 2e), which could imply that this cluster includes prion-vulnerable neurons that have undergone such extensive transcriptional changes that they no longer cluster together with their originating subclass. However, since we cannot link this mixed cluster of neurons with a specific subclass, we cannot validate whether their transcriptional changes are owing to prion biology, or are artefactual in nature. We felt it was more appropriate to focus on the well-defined neuronal subsets in the revised manuscript, and therefore removed the analysis on the 'Slc7a11' neuronal clusters.

- S100b, Il33, Trdn, and Cybrd1 were named as differentially expressed in prion disease in accordance with the literature; these do not appear in the current version, while Rbm3, Ttr, IEGs and others had not been described in the previous analysis.

Author response: These genes were differentially expressed in the 'Slc7a11' neurons, which, as mentioned above, we chose not to focus on in the revised manuscript because their origin is unclear. Furthermore, in the course of revising our analysis, we switched to a more robust method for differential expression analysis, using MAST's two-part hurdle model instead of Seurat's Wilcoxon rank-sum test. The MAST differential expression analysis was more conservative and identified fewer differentially expressed genes compared to the Seurat method (see Figure 1 below). Thus, S100b, Il33, Trdn, and Cybrd1 no longer met the criteria for differential expression. On the other hand, Rbm3, Ttr, and many of the IEGs were highly consistent between the MAST and Seurat differential expression methods (See Figure 2 below), and this is part of the reason we chose to focus on these genes and validate them using RNAscope.

Figure 1. Volcano plots show differentially expressed genes identified through Seurat’s Wilcoxon rank-sum test, and MAST’s two-part hurdle model.

Figure 2. Heatmap and violin plot shows genes that were consistently differentially expressed in the MAST and Seurat differential expression analysis.

It was especially important that the authors now explicitly name, display and discuss individual genes altered in prion disease - rather than just general signatures or processes.

The newly added validation of some of these genes by Scope, and against published bulk datasets, in Figure 4 is overall convincing. Additional clarification of the statistics in Fig. 4g will be helpful (I understand from the methods validation came from 3 mice; how many sections? how many cells?)

Author response: Due the costs associated with the RNAscope reagents, we analyzed one slide (with 3-4 coronal sections) per probe-panel per mouse. We provide the number of ROIs and number of cells used for each probe-panel/brain-region combo in Supplementary Figure 6. Typically, we aimed for at least 20 ROIs per brain region per mouse and analyzed at least 10,000 cells per region.

Finally, I appreciate the authors' clarifications in the text. For example, the first paragraph now openly describes the disease model, sampling approach, how cell type abundances are out of balance, and how neurons were integrated with an existing neuronal dataset.

Also the Scope validation of MSN/GABA depletion is now better explained and justified.

Changes in the method sections were important, this section is now much more clear.

Reviewer #3 (Remarks to the Author):

Author Response: We thank the reviewer for assessing our manuscript and for the constructive feedback.

Slota et al used single cell RNA Seq technology to analyse gene expression profiles of enriched cerebral neuron populations from CD1 mice infected with RML mouse-adapted scrapie prions, or mock infected control mice. They identified distinct gene expression responses in neuron subpopulations, and used level downregulation of synaptic gene expression to score neuronal damage and identified subsets of prion vulnerable neurons. Further, they compared gene expression profiles of prion-vulnerable and prion-resistant neurons and found that differences in baseline gene expression could influence neuronal vulnerability, highlighting increased expression of the neuroprotective cold shock protein Rbm3 in resistant neurons. Altered gene expression of selected candidates was validated using RNAscope analysis using brain sections of prion-infected and mock-infected mice. Moreover, comparison of their datasets to previously published transcriptomes from Alzheimer's disease models revealed that transcripts related to vulnerability overlap in Alzheimer's and prion disease models.

The manuscript has been peer-reviewed by two experts and this reviewer was asked to comment on the authors responses to previous concerns.

Reviewer 1:

(1) Data quality and comprehensiveness

The authors sequence >130k cells extracted from 'cerebral' brain tissue after depletion of non- neurons, and >100k cells meet relevant quality criteria. Out of these 100k cells, only a small minority is neuronal (Fig. 1a); and in fact, the remaining analysis is based on <9k cells with mature neuronal signatures. This ratio is far below what is expected – even considering that ex vivo stress during dissociation

disproportionally affects neurons. I acknowledge that from this particular tissue it may be difficult to keep neurons viable, also considering the mice' likely advanced age at the time of sampling (although ages are not provided). Yet, even without neuronal enrichment, I would not expect more than 2x more non-neuronal than neuronal cells in the cerebrum, perhaps 3x in the striatum. The relative high number of endothelial cells is particularly unusual.

In the same way, it is very unexpected that the authors would detect so many neurons with immature signatures. The immature neuron population looks like it is derived from the SVZ/RMS, but appears substantially overrepresented, again pointing towards unfavorable dissociation conditions for mature neurons, compared to the less vulnerable cell types.

The remaining dataset of ~9k neurons that Slota et al. analyze are consistently matched and compared between a relevant control and the disease state, certainly making those comparisons valid in principle. However, given the massive neuronal loss, it appears likely that many sensitive neurons and neuronal populations were selectively affected, questioning the comprehensiveness of the dataset.

Response: Isolation of intact live neurons from prion infected mice was indeed challenging, and we suspect that endothelial cells and immature neurons were not effectively depleted. The bead based neuronal enrichment kit from Miltenyi Biotech utilizes a proprietary mixture of antibodies that target non-neuronal cell surface markers. It seems likely that the antibody mixture was ineffective at removing endothelial cells and immature neurons in our case, which would explain the abundance of these cell types in our dataset.

We agree that the dataset lacked comprehensiveness, particularly in the case of thalamic/hypothalamic neurons that were absent from our dataset. Even the high resolution Allen Brain atlas reference atlas was supplemented with single nucleus RNAseq data for challenging- to-isolate neurons from the midbrain and hindbrain (Yao et al. 2023). This is a limitation that we were unfortunately not able to get around. In our case, cell isolation was performed on prion infected brain tissues within a containment suite with limited available equipment. Crucially, neither an ultracentrifuge nor a fluorescent cell sorter were available, which are necessary for single-nucleus based approaches. Thus, our only option was to isolate live neurons from prion- infected mice at an advanced age (mean ~210 days). We believe however, that this has resulted in a unique dataset with higher quality gene expression measurements compared to the single- nucleus datasets that are frequently used to study neuronal vulnerability in other neurodegenerative disorders (e.g. ref's 36-38 in the revised manuscript). The quality of the gene expression measurements and differential expression analyses is backed by our RNA-scope based validation of prion-altered transcripts that we have now included in the manuscript (Figure 4). We are not aware of any other studies that have used live single cell RNA sequencing in the study of neuronal toxicity or vulnerability, making our dataset a unique resource for the neuroscience community.

R3: This is a major concern that has also been raised by reviewer 2, and a challenge which the authors were not able fully address because it would essentially mean to repeat the entire experiment. However, despite the technical challenges the authors provide a highly relevant and unique dataset that enabled them to identify vulnerable populations of neurons, and baseline gene expression profiles that are related to resistance to prion-induced damage. Due to the long incubation time of prion disease in mice, naturally the disease models are at an advanced/old age at the time they reach clinical disease, which poses an additional challenge for isolating live neurons. However, the experiment is well controlled by using age matched mock-infected control mice for isolation of neurons according to the

same protocol used for prion-infected mice to rule out gene expression changes induced by the relatively long dissociation time.

(2) Cell type analysis

Slota et al. find 21 neuronal cell types in their analysis, and annotate them as GABA, GLUT or MSN, and 3-4 marker genes. However, there is no attempt to map these cell types with respect to existing atlases (Allen Mouse Brain Yao et al. 2023, Linnarsson datasets, etc.). For example, there was no mention of what the Slc7a11-neurons that are analyzed in more detail might be – and Slc7a11 is a marker much more expressed in the vasculature (VLMC, Pericytes, perhaps astrocytes).

Further, in scoring the cell types according to vulnerability, the authors later state that there is no relation between vulnerability and neuronal subtype (and I believe they refer to classes, GABA, GLUT or MSN). For lack of a comprehensive attempt to understand which subtypes they describe this statement becomes less convincing.

Response: Thank you for this suggestion. We have now annotated cell types based on mapping cells to the Allen Brain Atlas's high-resolution mouse brain cell atlas (Yao et al. 2023). The major neuronal classes captured by our dataset corresponded to cortical, hippocampal, and anterior olfactory nucleus glutamatergic neurons, nucleus accumbens GABAergic neurons, striatal medium spiny neurons, with a smaller number of Pvalb+, Sst+, Sncg+, and Lamp5+ GABAergic neurons.

R3: This response is satisfactory.

(3) Data presentation

The manuscript would greatly benefit from scaling down on presenting lots of 'compressed' data; i.e. gene modules, scores, processes, unannotated heatmaps. Instead, at least examples of more 'raw' data would help interpret the data in a more nuanced way, such as to be able to judge effect sizes, and significance. The authors don't show many examples of genes, and they are often hidden in very dense heatmaps. Instead, they could choose to show differential expression of genes in a specific cell type as volcano plot or similar. There is not much mention of specific genes until the Discussion, and the genes highlighted in the Discussion are not shown in the data (main figures or supplementary).

Response: Thank you for this insight. It can be challenging to choose appropriate data visualizations for a dataset of this complexity. In the course of revising this manuscript, we have extensively revised the figures to include more examples of raw data through volcano plots and violin plots etc.

R3: This reviewer is unable to assess how much data presentation has improved as they have not seen the original manuscript. Current data presentation is appropriate.

(4) Validation

Slota et al. performed RNA Scope to quantify canonical cell classes and their expression intensity. This validation feels almost unrelated to the scRNA-seq dataset, as the authors do not describe that the canonical cell types or markers are depleted outside the RNA Scope analysis. It would be more relevant to focus on a few specific cell types, e.g. the Slc7a11 population, validate where it is located, and how the expression of 1-2 highly DEGs is affected over the course of the disease.

Response: As mentioned in our response to your first comment, our live scRNAseq dataset is not well suited to detect changes in cell composition. Thus, we were not surprised that changes in cell type abundance were not reflected. However, in figure 6 we demonstrate that the depletion of medium spiny neurons in the RNA scope analysis is consistent with them being identified as especially vulnerable to prion infection in our scRNAseq analysis.

We have now conducted a second RNA scope analysis to validate the prion-altered expression of four promising genes – Ttr, Rbm3, Egr1, and Rab6b. These data are presented in figure 4, and we detected robust prion-alterations of Rbm3, Egr1, and Rab6b. We would not expect to see much in the way of temporal gene alterations over the course of the disease, since neuronal loss/dysfunction is usually only detected and the experimental endpoint in murine models of prion disease. Instead, we demonstrated that these genes are prion-altered only near the endpoint of prion disease using previously published bulk RNAseq datasets that longitudinally examined prion infection. These late changes in neuronal gene expression are not surprising since, due to animal welfare regulations, prion-infected mice are sacrificed at relatively early disease stages when neuronal loss is minimal.

R3: The authors have appropriately addressed the concerns.

As general good publication practice, I would expect better description of the methodology throughout the main text. For example:

- No age of disease induction (or sacrifice) is provided.

Response: We have included these details in the methods section. Mice were inoculated at 7-weeks old, and were sacrificed between ~150-190 days post inoculation, meaning they were an average age of ~210 days old at the time of sacrifice. The RML mouse model we employed is extensively used within the prion field, and is essentially a standard model. This is why only minimal details are included in the main text.

- There is no mention of the dissection strategy in the main text, in Methods the authors write ‘cerebral’ tissues were collected without detailing which precise structures were included in the data. (Here, it might also be useful to see histological comparisons of how the ‘cerebral’ brain tissue looks at clinical endpoint, vs. mock injected, justifying the dissection strategy.)

Response: We apologize for the confusion with respect to the brain tissues that were used. We essentially used the whole brain, but removed the cerebellum and hindbrain to reduce sample complexity, and because relatively few pathological changes are detected within the cerebellum in this RML mouse model. Furthermore, the olfactory bulb was left behind when brains were removed. Thus, the remaining brain tissue after dissection corresponded to the entire forebrain + midbrain. This mouse model has been extensively examined in previous histological studies. For a particularly comprehensive histological study, see: <https://doi.org/10.1177/0300985819861708>.

- Analysis methods are barely described in the text at all. Without going into details, I expect an outline / intuition of the authors’ intent in any new analysis when following the main text. An extreme example of this are the different analyses compared in Figure 7c-d.

Response: The revised manuscript includes additional methodological details in the main text. This has made the manuscript lengthier, but we hope the robustness of the dataset and analysis are now easier to judge.

R3: The authors have added more details as requested.

Reviewer #2:

1. The cell isolation protocol is flawed. Cells were dissociated for 40 min at 37 °C which result in artifactual changes in gene expression patterns as reported by multiple publications (<https://www.ncbi.nlm.nih.gov/pmc/articles/PMC5665481/> ; <https://genomebiology.biomedcentral.com/articles/10.1186/s13059-019-1830-0>).

Response: We agree that the extended dissociation step was a limitation of our protocol for cell isolation. See our responses to reviewer # 1 for a more detailed explanation of why we chose to isolate live neurons from prion infected mice. However, cells were isolated from both prion- and mock- infected mice according to the same protocol, enabling comparisons between the two groups.

Furthermore, we have now validated several prion-altered transcripts using RNA scope (Figure 4). This confirms the robustness of the dataset and verifies that the observed gene expression changes are 'real', rather than artefactual.

R3: See response to reviewer 1 comment.

2. The authors used vulnerability score cutoff of 0.5 to determine whether a neuronal population is vulnerable or resistant for their dataset. They adjusted vulnerability score thresholds "to 1.14 and 0.3 for the Grubman et al. and Leng et al. datasets respectively." The purpose is "to ensure a comparable number of vulnerable and resistant neurons for analysis". All the analysis criteria were arbitrary without any scientific justification.

Response: For identification of vulnerability-correlated transcripts, we have now performed a differential expression analysis based on the quantitative vulnerability score value, instead of using an arbitrary classification of neurons as either 'resistant' or 'vulnerable' to group cells for differential expression testing. The classification as either 'resistant' or 'vulnerable' is now only used for visualization purposes, and we now have adjusted the vulnerability score cut-off to 1 because this is equivalent to a synapse-gene module depletion with FDR = 0.1. In other words, neurons are classified as vulnerable if they display a depletion of synapse-gene expression that reaches (or nearly reaches) statistical significance.

R3: The authors have adjusted their analysis and provide sufficient justification for their approach.

3. Fig. 1b: no distinguishing marker for epen.15, vlmc.26, npc.30, peri.31, or oligo.33 cell type is shown.

Response: To show expression of more marker genes, Figure 1b displays expression of numerous marker genes as a heatmap, and Supplementary Figure 1 demonstrates marker gene expression for each major brain cell type in a violin plot.

R3: Heat maps and additional violin plots as shown in main and supplementary figure are satisfactory.

4. Fig. 1: endothelial cells were divided into 10 clusters but no explanation of genes that distinguish different subpopulations.

Response: Endothelial cells are not the focus of our investigation, which is why we have not went to the extent of defining the different clusters in this manuscript. However, we have now provided genes that globally distinguish cell clusters in the full scRNAseq atlas as supplementary data file 2.

R3: The argument that endothelial cells are not the focus of their study is justified.

5. Mouse cortex neuronal cell types have been very well characterized with detailed layer- specific neuronal subtypes and marker genes. This study did not reference or identify any of these well-defined neuronal subtypes. It is unclear how credible their cell type classification is.

Response: We have improved our cell classification by mapping cells to the Allen brain atlas's high-resolution mouse brain cell atlas. See our response to reviewer #1 for further details.

R3: The improved cell classification addresses the reviewer's concern.

6. The authors identified prion-associated composition changes in glutamatergic and GABAergic neurons, but not in medium spiny neurons. However, in their analysis, msn.neu was assigned as the reference for the comparison. scCODA method used in the manuscript requires a reference to be able to identify compositional changes. scCODA can automatically select an appropriate cell type as the reference or uses a pre-specified reference cell type. It's unclear why the authors did not use the automatically reference selection method. Why msn.neu was used as the reference and how selection of reference affect the result and conclusion?

Thank you for this suggestion. We have now employed the automated reference selection method in scCODA. The specific cell clusters that were selected as reference in each cell composition analysis are provided in Supplementary data File 3.

R3: The authors followed the reviewer's suggestion.

7. "The five RML-infected mice used for single-cell RNA sequencing reached the clinical endpoint at 164, 172, 178, 179, and 182 days post infection (dpi), while the four Mock-infected mice were sacrificed at 148, 168, 185, and 189 dpi." Each mice has a different dpi, why? The Mock-infected mice were collected at a different time point than their treatment group animals. This seems to lost the meaning of serving as the "control" group.

Response: Prion-infected mice were sacrificed based on when they exhibit clinical signs of disease, rather than being sacrificed at a specific timepoint. This is why each mouse has a different dpi.

Furthermore, our cell isolation protocol could be applied only to one mouse per day. Therefore, the Mock treated mice were sacrificed on different days than the prion infected mice. We chose an age range of mock mice to match the prion infected mice, and this is standard practice for murine models of prion disease.

R3: This reviewer agrees with the authors. The age range of mock-infected mice is appropriate.

8. The authors performed pseudotime analysis on glut.neu.3, glut.neu.5, and glut.neu.10 subpopulations. Initially these clusters were described as distinct neuronal subpopulations. Later the

authors suggested that they represented a glutamatergic neuronal subset undergoing disease-associated transcriptional changes. These are conflicting results without experimental evidence to support either. If they are distinct neuronal subpopulations it does not make sense to perform pseudotime analysis on these subpopulations. The pseudotime DEGs just represent genes differentially expressed between different subpopulations.

Response: Thank you for pointing this out, we apologize for this oversight. The pseudotime analysis has been removed from the revised manuscript because these cell clusters were annotated as 'mixed' when we applied the improved cell classification system.

R3: This is satisfactory.

9. Figure 5C: The authors quantified signal intensity and compared gene expression differences using individual neurons. It is unclear how many mice were sampled. The statistics should be calculated using individual mice instead of using individual cells.

Response: We used a separate set of three prion-infected and three mock-treated mice for RNAscope analysis. As mentioned, we have now included a second RNAscope dataset to validate prion-altered gene expression. We have also improved our image analysis pipeline by quantifying gene expression as puncti density per mm² rather than signal intensity. For these gene expression measurements, we took the per-mouse averages for statistics.

R3: The RNAscope analysis performed by the authors is highly relevant to confirm changes in gene expression, as it supports the argument of prion-induced gene expression changes rather than artificial changes induced during isolation and dissociation of neurons. The authors have further refined their analysis following the reviewer's suggestions.

10. The logic of the statement "Notably, prion-vulnerable neurons highly expressed the metabotropic glutamate receptors Grm5 and Grm8, known mediators of prion toxicity^{58,59}, while prion-resistant neurons expressed Gal3st160 and Stmn2, implicated as genetic risk loci for prion disease. These findings suggest that prion-vulnerable neurons tend to be more terminally differentiated and excitable, whereas resistant neurons are less mature and express genes associated with axon regeneration." is not clear.

Response: We apologize for the confusion. This section of the manuscript has been revised and now reads as follows:

"We identified biological processes that were enriched with the list of transcripts differentially expressed by vulnerable and resistant neuronal subsets using Enrichr (Figure 7e). Prion-vulnerable neurons highly expressed transcripts linked to synaptic transmission, memory, and cellular adhesion, while resistant neurons abundantly expressed genes related to vesicle trafficking, negative regulation of supramolecular fiber assembly, and ubiquitin-mediated protein catabolism. Interestingly, several vulnerability-correlated transcripts have been previously linked with prion disease. For instance, the metabotropic glutamate receptors Grm5 and Grm8 mediate prion toxicity^{46,47}, while Bmerb148, MacroD247 and Stmn249 are implicated as putative genetic risk loci for prion disease. Several tubulin-encoding genes, including Tubb2a, Tubb2b, Tubb4a, Tubb4b, and Tubb5, were highly expressed by prion-resistant neurons. This is of note because in a previous study we found PrPC-overexpressing neuronal cultures, which were demarked by high tubulin expression, to completely resist prion infection⁵⁰. Moreover, the baseline expression of the neuroprotective stress-response gene Rbm337³⁸

was higher in prion-resistant neurons (Supplementary File 4), and this was supported by RNA FISH that detected conspicuously low levels of *Rbm3* expressed by selectively vulnerable medium spiny neurons (Figure 4g). The prion-associated upregulation of *Rbm3* consistently seen across different neuronal subsets (e.g. Figure 3 and Figure 4) could imply a protective response. Thus, it is tempting to speculate that neuronal expression of specific genes could modulate vulnerability to prion-induced damage.”

R3: This is a satisfactory response.

11. The authors found Galnt13 as the only vulnerability marker common to all four datasets. It is unclear whether this gene has anything to do with neuron vulnerabilities.

Response: Galnt13 catalyzes the O-linked glycosylation of mucins. We thought this was interesting because mucins have been shown to impair prion misfolding in cell-free conversion assays, presenting an analytical barrier for detecting prions in saliva, which contains mucins (PMID 29950332).

Galnt13 likely plays a different role in the brain, for instance, it may promote neurogenesis through glycosylation and stabilization of PDPN (PMID 12407114). Glycosylation is seemingly linked to prion replication, since heparin (PMID 24648544) and dextran sulfate (PMID 20957174) can serve as co-factors that modulate prion replication. Thus, it is plausible that the modulation of glycosylation by Galnt13 could influence prion replication, which could have a downstream effect on neuronal vulnerability. However, since we do not have strong data to support this hypothesis, the reference to Galnt13 has been removed from the text.

R3: This is indeed an interesting link that the authors might want to investigate in more detail in a follow up study. Removing the reference to Galnt13 is appropriate at this point.

Additional minor comments R3:

- Figure 1 legend: volcano plot is panel d (not c), hierarchical clustering is panel e (not d)

Author response: Thank you for pointing this out. This has been corrected.

- in the discussion, the authors could provide further arguments for the use of live neurons rather than single nuclei RNAseq, as done in previous studies, that avoids some of the challenges faced with neuron enrichment. Did they identify transcripts that would have been lost if they had used snRNA-Seq?

Author response: This is an interesting question, but it is difficult to answer precisely without directly comparing a single cell and single nucleus dataset made from the same tissue. We certainly suspect that the use of live neurons resulted in enhanced detection of additional genes that might be missed with snRNAseq. In support of this notion, more genes were detected in the MAST differential expression analysis (which removes genes detected in < 10% of cells) of our live single cell RNAseq dataset compared to the Alzheimer’s disease snRNAseq datasets that we analyzed as part of this study. Notably, *RBM3* was not detected in the AD snRNAseq datasets, possibly owing to the lack of cellular mRNAs captured by snRNAseq. Indeed, the expression level of *Rbm3* appeared to be quite low in our scRNAseq dataset (Figure 3f), yet it was robustly detected via RNA scope (Figure 4b,e,g). We now refer to this point in the discussion section at lines 382-385:

“We did not detect neuronal *RBM3* expression in the Leng or Grubman Alzheimer’s disease datasets, possibly because it is more challenging to detect low-abundance transcripts with single-nucleus RNA

sequencing. Thus, we were unable to conclude whether *Rbm3*'s association with neuronal vulnerability is common to prion and Alzheimer's disease."

- while such extensive scRNASeq studies cannot be performed in parallel with multiple prion strains, it is recommended to discuss the lack of validation with respect to different strains

Author response: This is a good point. We have added the following to the final paragraph of the discussion:

"As this study employed only one strain of mouse-adapted scrapie, analyzing and comparing neuronal vulnerability between different prion strains would strengthen our findings."